# Unsupervised Disentanglement of Content and Style via Variance-Invariance Constraints

**Yuxuan Wu**[♮]  **Ziyu Wang**[♯♭]  **Bhiksha Raj**[♮♭]  **Gus Xia**[♭♯]

[♭] Mohamed bin Zayed University of Artificial Intelligence
[♯] New York University Shanghai
[♮] Carnegie Mellon University

## Abstract

We contribute an unsupervised method that effectively learns disentangled *content* and *style* representations from sequences of observations. Unlike most disentanglement algorithms that rely on domain-specific labels or knowledge, our method is based on the insight of domain-general statistical differences between content and style — content varies more among different fragments within a sample but maintains an invariant vocabulary across data samples, whereas style remains relatively invariant within a sample but exhibits more significant variation across different samples. We integrate such inductive bias into an encoder-decoder architecture and name our method after V3 (**v**ariance-**v**ersus-in**v**ariance). Experimental results show that V3 generalizes across multiple domains and modalities, successfully learning disentangled content and style representations, such as pitch and timbre from music audio, digit and color from images of hand-written digits, and action and character appearance from simple animations. V3 demonstrates strong disentanglement performance compared to existing unsupervised methods, along with superior out-of-distribution generalization under few-shot adaptation compared to supervised counterparts. Lastly, symbolic-level interpretability emerges in the learned content codebook, forging a near one-to-one alignment between machine representation and human knowledge.[1]

## 1 Introduction

Learning abstract concepts is an essential part of human intelligence. Even without any label supervision, we humans can abstract rich observations with great variety into a category, and such capability generalizes across different domains and modalities. For example, we can effortlessly perceive a picture of a "cat" captured at any angle or set against any background, we can perceive the symbolic number "8" from an image irrespective of its color or writing style variations, and we can perceive an abstract pitch class "A" from an acoustic signal regardless of its timbre. These concepts form the fundamental vocabulary of our languages—be they natural, mathematical, or musical—and underpin effective and interpretable communication in everyday life.

Our goal is to emulate such abstraction capability using machine learning. We choose a *content-style representation disentanglement* approach as we believe that representation disentanglement offers a more complete picture of abstraction—concepts that matter more in communication, such as an "8" in a written phone number or a note pitch "A" in a folk song, are usually perceived as *content*, while the associated variations that often matter less in context, such as the written style of a digit or the singing style of a song, are perceived as *style*. In addition, content is usually symbolized and associated with rigid labels, as we need precise control over it during communication. E.g., to write "8" as "9" in a phone number or to sing an "A" as "B" in a performance can be a fatal error. In comparison, though style can also be described discretely, such as an "italic" writing or a "tenor" voice, a variation over it is usually much more tolerable.

---

[1]Code is available at `https://github.com/Irislucent/variance-versus-invariance`. Demo can be found at `https://v3-content-style.github.io/V3-demo/`.

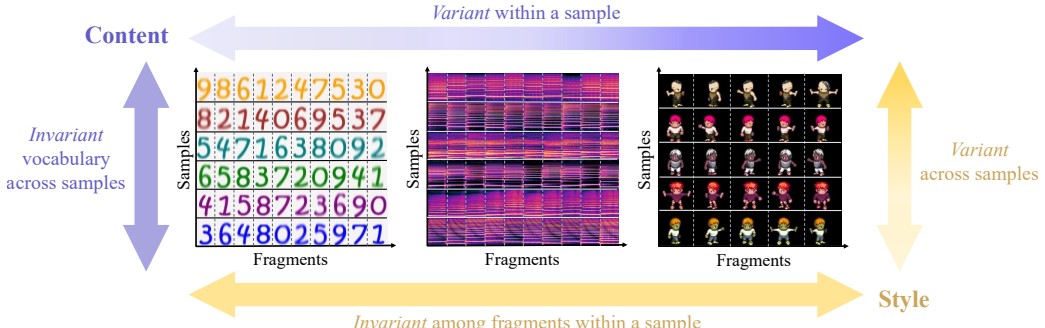

Figure 1: An illustration of the variance-versus-invariance constraints in *content* and *style*. Here, *content* refers to the symbols. Each row represents a data sample, which is divided into multiple fragments along columns. Each fragment contains one content-style pair. For example, digits (e.g., 9, 8, 6) can represent *content*, while colors (e.g., orange, brown, teal) represent *style*.

In the machine learning literature, significant progress has recently been made in content-style disentanglement for various tasks, including disentangling objects from backgrounds (Hong et al., 2023), characters from fonts (Liu et al., 2018; Xie et al., 2021), pitch from timbre (Lu et al., 2019; Bonnici et al., 2022), and phonemes from speaker identity (Qian et al., 2019; Li et al., 2021). However, most existing models are either limited to specific domains (Yingzhen and Mandt, 2018; Bai et al., 2021; Luo et al., 2022) or rely heavily on domain-specific knowledge as implicit supervision. The supervision forms can be explicit content or style labels (Liu et al., 2017; Zhu et al., 2017; Park et al., 2020; Karras et al., 2019; Choi et al., 2020; Bonnici et al., 2022; Patashnik et al., 2021; Kwon and Ye, 2022), pre-trained content or style representations (Qian et al., 2020b; 2019), or paired data showcasing the same content rendered in different styles or vice versa (Isola et al., 2017; Sangkloy et al., 2017). In addition, the disentangled representations often fell short in generalizing to new contents or styles, and they lack interpretability at a symbolic level and do not align well with human perceptions (Zhang et al., 2021; Nauta et al., 2023).

To address the aforementioned challenges, achieving more generalizable and interpretable disentanglement in an unsupervised manner, we introduce V3 (*variance-versus-invariance*). V3 disentangles content and style by leveraging meta-level prior knowledge about their inherent statistical differences. As shown in Figure 1, our design principle is based on the observation that **content and style display distinct patterns of variation**—*content undergoes frequent changes within different fragments of a sample yet maintains a consistent vocabulary across data samples, whereas style remains relatively stable within a sample but exhibits more significant variation across different samples*.

In this paper, we adopt the vector-quantized autoencoder architecture and incorporate variance-versus-invariance constraints to guide the learning of latent representations that capture content-style distinctions. We demonstrate that V3 effectively generalizes across distinct areas: disentangling pitches and timbres from musical data, disentangling numbers and ink colors from images of digits, and disentangling character actions and appearances from game video clips. Experimental results show that our approach achieves more robust content-style disentanglement than unsupervised baselines, and outperforms even supervised methods in out-of-distribution (OOD) generalization and few-shot learning for discriminative tasks. Lastly, symbolic-level interpretability emerges with a near one-to-one alignment between the vector-quantized codebook and human knowledge, an outcome not yet seen in previous studies. In summary, our contributions are as follows:

- **Unsupervised content-style disentanglement:** We introduce V3, an unsupervised method leveraging meta-level inductive bias to disentangle content and style representations, without requiring paired data, content or style labels, or domain-specific assumptions.

- **Out-of-distribution generalization:** As a result of successful content-style disentanglement, V3 shows better out-of-distribution generalization capabilities compared to supervised methods in few-shot settings, that is, recognizing content when presented with only a few examples of unseen styles.

- **Emergence of interpretable symbols:** Given the availability of semantic segmentations, V3 can foster the development of interpretable content symbols that closely align with human knowledge.

## 2 RELATED WORK

The content-style disentanglement as well as the related style transfer problem has been well explored in computer vision, especially in the context of image-to-image translation. Early works mostly require paired data of the same content with different styles (Isola et al., 2017; Sangkloy et al., 2017), until the introduction of domain transfer networks that can learn style transfer functions without paired data (Zhu et al., 2017; Liu et al., 2017; Taigman et al., 2016; Bousmalis et al., 2017; Park et al., 2020; Choi et al., 2018; 2020; Karras et al., 2019; Xie et al., 2022a;b). Although these methods are unsupervised in the sense that they do not require paired data, they still require concrete labels of styles to identify source and target domains, and there are no fully interpretable representations of either content or style.

A similar trajectory of research has also been followed in other domains including speech Qian et al. (2019); Kameoka et al. (2018); Kaneko et al. (2019); Wu et al. (2023) and music Lu et al. (2019); Bonnici et al. (2022); Luo et al. (2022); Lin et al. (2023); Zhang et al. (2024); Lin et al. (2021). To mitigate the requirement for supervision, some methods utilize domain-specific knowledge and have achieved better disentanglement results, including X-vectors of speakers Qian et al. (2019; 2020a), the close relation between fundamental frequency and content in audio Qian et al. (2020a;b), or pre-defined style or content representations Yang et al. (2019); Wang et al. (2020; 2022).

Pure unsupervised learning for content and style disentanglement has not been well explored. Notable attempts include adversarial training-based methods (Chen et al., 2016; Ren et al., 2021), mutual information neural estimation (MINE) (Belghazi et al., 2018; Poole et al., 2019; Tjandra et al., 2020a; Zhang and Dixon, 2023), and low-dimensional representation learning with physical symmetry (Liu et al., 2023). But these methods often suffer from the training stability issue or have not been proven to be domain-general. Other unsupervised methods on sequential data, such as Disentangled Sequential Autoencoder (DSAE) and its variants, leverage the nature of content and style to learn their representations at different scales, but their applications are limited to purely sequential data with a static style (Hsu et al., 2017; Yingzhen and Mandt, 2018; Bai et al., 2021; Luo et al., 2022; 2024; Yin et al., 2022).

A technique often associated with learned content is vector quantization (VQ) (Van Den Oord et al., 2017). Recent efforts have built language models on top of VQ codes for long-term generation, indicating the association between VQ codebook and the underlying information content (Yan et al., 2021; Tan et al., 2021; Copet et al., 2024; Garcia et al., 2023; Tjandra et al., 2020a;b; Vali and Bäckström, 2023). A noticeable characteristic of these studies is the use of large codebooks, which limits the interpretability of representations. We borrow the idea of a *small* codebook size from categorical representations (Chen et al., 2016; Ji et al., 2019), targeting a more concise and unified content code across different styles, while keeping the high-dimensional nature of VQ representations.

## 3 METHODOLOGY

Considering a dataset consisting of $N$ data samples, where each sample contains $L$ fragments, we aim to learn each fragment's *content* and *style* representation with the inductive bias illustrated in Figure 1. Intuitively, the fragments within each data sample have a relatively frequently-changing content and a relatively stable style. For different data samples, their style exhibits significant variations and their content more or less keeps a consistent vocabulary. In the following, we first introduce the autoencoder architecture V3 is built upon, then the variability statistics to quantify the changing patterns of content and style, and the proposed variance-versus-invariance constraints.

### 3.1 MODEL ARCHITECTURE

The model architecture of V3 is illustrated in Figure 2. Let $\boldsymbol{X} = \{\boldsymbol{x}_{ij}\}_{N \times L}$ be the dataset, where $\boldsymbol{x}_{ij}$ corresponds to the $j$-th fragment of the $i$-th sample. We use an autoencoder architecture to learn the representations of $\boldsymbol{x}_{ij}$. The encoder encodes the input data $\boldsymbol{x}_{ij}$ to the latent space, which is split

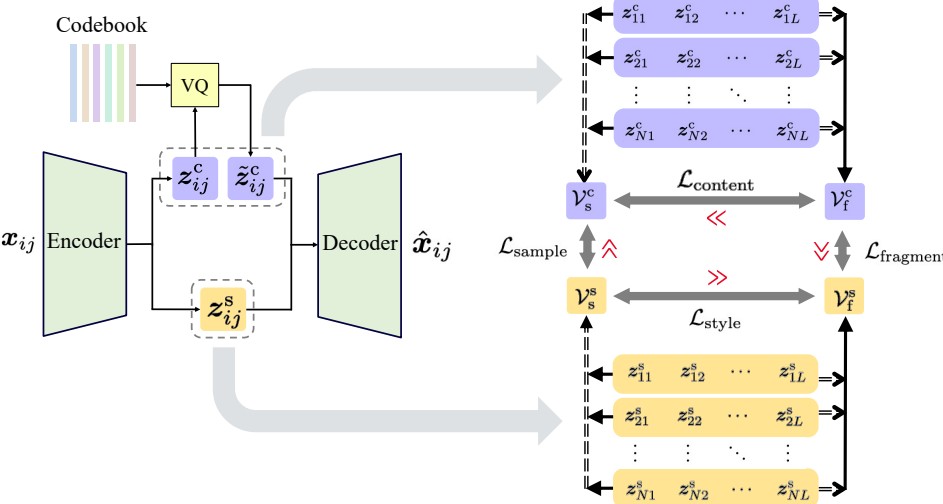

Figure 2: The model architecture of V3. Left: The autoencoder has two branches for content and style respectively, where the content branch has a VQ layer at the encoder output. Right: the V3 constraints, where double-dashed arrows represent measuring the variability by $\nu_k(\cdot)$, and solid arrows represent taking the average.

into to $z_{ij}^{\text{c}}$ and $z_{ij}^{\text{s}}$. We use vector quantization as the dictionary learning method for content. Every content representation $z_{ij}^{\text{c}}$ is quantized to the nearest atom in a codebook of size $K$ as $\tilde{z}_{ij}^{\text{c}}$. The decoder integrates $\tilde{z}_{ij}^{\text{c}}$ and $z_{ij}^{\text{s}}$ and reconstructs the fragment $\hat{x}_{ij}$. The overall loss function is the weighted sum of three terms:

$$\mathcal{L} = \mathcal{L}_{\text{rec}} + \alpha \mathcal{L}_{\text{vq}} + \beta \mathcal{L}_{\text{V3}}. \tag{1}$$

Here, $\mathcal{L}_{\text{rec}}$ is the reconstruction loss of $X$ and $\mathcal{L}_{\text{vq}}$ is the VQ commit loss (Van Den Oord et al., 2017):

$$\mathcal{L}_{\text{rec}} = \frac{1}{N \times L} \sum_{i=1}^{N} \sum_{j=1}^{L} \|x_{ij} - \hat{x}_{ij}\|_2, \tag{2}$$

$$\mathcal{L}_{\text{vq}} = \frac{1}{N \times L} \sum_{i=1}^{N} \sum_{j=1}^{L} \|z_{ij}^{\text{c}} - \text{sg}(\tilde{z}_{ij}^{\text{c}})\|_2, \tag{3}$$

where $\text{sg}(\cdot)$ is the stop gradient operation of the straight-through optimization. The final term $\mathcal{L}_{\text{V3}}$ is the proposed regularization method to ensure unsupervised content-style disentanglement, which we introduce in the rest part of this section. (For more details of the model architecture and data representations, we refer the readers to Appendix B.)

## 3.2 VARIABILITY STATISTICS

We define four statistics to measure the *degree of variability* in accordance with the four edges of Figure 1. These statistics are based on a backbone variability measurement $\nu_k(\cdot)$, where $k$ represents the dimension along which variability is computed. In this paper, we define $\nu_k(\cdot)$ as the mean pairwise distance (MPD). Formally, for a vector $z$ of length $D$,

$$\nu_{i=1}^{D}(z_i) := \text{MPD}_{i=1}^{D}(z_i) = \frac{1}{D(D-1)} \sum_{i=1}^{D} \sum_{j=1, j \neq i}^{D} \|z_i - z_j\|_2. \tag{4}$$

The motivation for using MPD is that it is more sensitive to multi-peak distributions than standard deviation, which is preferred when learning diverse content symbols in a sample. We compare different choices of $\nu_k(\cdot)$ in Appendix D.

**Content variability within a sample** ($\mathcal{V}_f^c$). We first compute the variability of content along the fragment axis and take the average along the sample axis. The value is the average of content codes before and after vector quantization:

$$\mathcal{V}_f^c = \frac{1}{2N} \sum_{i=1}^{N} \nu_{j=1}^{L}(\boldsymbol{z}_{ij}^c) + \frac{1}{2N} \sum_{i=1}^{N} \nu_{j=1}^{L}(\tilde{\boldsymbol{z}}_{ij}^c). \tag{5}$$

**Content variability across samples** ($\mathcal{V}_s^c$). Theoretically, we aim to measure the consistency of codebook usage distribution along the sample axis, which is not differentiable. In practice, we compute the center of the content code along the fragment axis and measure the variability of the centers along the sample axis. It serves as a proxy of codebook utilization. Also, we consider both content codes before and after vector quantization:

$$\mathcal{V}_s^c = \frac{1}{2} \nu_{i=1}^{N}\Big(\frac{1}{L} \sum_{j=1}^{L} \boldsymbol{z}_{ij}^c\Big) + \frac{1}{2} \nu_{i=1}^{N}\Big(\frac{1}{L} \sum_{j=1}^{L} \tilde{\boldsymbol{z}}_{ij}^c\Big). \tag{6}$$

**Style variability within a sample** ($\mathcal{V}_f^s$). We compute the variability of style representations among fragments and take its mean across all samples:

$$\mathcal{V}_f^s = \frac{1}{N} \sum_{i=1}^{N} \nu_{j=1}^{L}(\boldsymbol{z}_{ij}^s). \tag{7}$$

**Style variability across samples** ($\mathcal{V}_s^s$). We compute the average style representation along the fragment axis and measure its variability along the sample axis:

$$\mathcal{V}_s^s = \nu_{i=1}^{N}\Big(\frac{1}{L} \sum_{j=1}^{L} \boldsymbol{z}_{ij}^s\Big). \tag{8}$$

### 3.3 Variance-Versus-Invariance (V3) Constraints

With the variability statistics, we can formalize the general relationship between content and style along the sample or fragment axis:

- Content should be more variable within samples than the aggregated content across samples, i.e., $\mathcal{V}_f^c \gg \mathcal{V}_s^c$.
- Style should be more variable across samples than within samples, i.e., $\mathcal{V}_s^s \gg \mathcal{V}_s^f$.
- Within a sample, content should be more variable than style, i.e, $\mathcal{V}_f^c \gg \mathcal{V}_f^s$.
- Across samples, style should be more variable than content, i.e., $\mathcal{V}_s^s \gg \mathcal{V}_s^c$.

We quantify the above contrasts as regularization terms, using the hinge function to cut off gradient back-propagation when the ratio between two variability statistics reaches a certain threshold $r > 1$, which stands for relativity (Bardes et al., 2022):

$$\mathcal{L}_{\text{content}} = \max(0, 1 - \frac{\mathcal{V}_f^c}{r \cdot \mathcal{V}_s^c}), \quad (\mathcal{V}_f^c \gg \mathcal{V}_s^c) \tag{9}$$

$$\mathcal{L}_{\text{style}} = \max(0, 1 - \frac{\mathcal{V}_s^s}{r \cdot \mathcal{V}_f^s}), \quad (\mathcal{V}_s^s \gg \mathcal{V}_f^s) \tag{10}$$

$$\mathcal{L}_{\text{fragment}} = \max(0, 1 - \frac{\mathcal{V}_f^c}{r \cdot \mathcal{V}_f^s}), \quad (\mathcal{V}_f^c \gg \mathcal{V}_f^s) \tag{11}$$

$$\mathcal{L}_{\text{sample}} = \max(0, 1 - \frac{\mathcal{V}_s^s}{r \cdot \mathcal{V}_s^c}). \quad (\mathcal{V}_s^s \gg \mathcal{V}_s^c) \tag{12}$$

We obtain the V3 regularization term (used in Equation 1) by summing up the four terms:

$$\mathcal{L}_{\text{V3}} = \mathcal{L}_{\text{content}} + \mathcal{L}_{\text{style}} + \mathcal{L}_{\text{fragment}} + \mathcal{L}_{\text{sample}}. \tag{13}$$

Table 1: Evaluation of digit and color disentanglement on PhoneNums using latent retrieval. Values are reported in percentage.

| Method | $K$ | Content | | | | Style | | | |
|---|---|---|---|---|---|---|---|---|---|
| | | PR-AUC | | Best F1 | | PR-AUC | | Best F1 | |
| | | $z^c \uparrow$ | $z^s \downarrow$ | $z^c \uparrow$ | $z^s \downarrow$ | $z^c \downarrow$ | $z^s \uparrow$ | $z^c \downarrow$ | $z^s \uparrow$ |
| V3 | 10 | 83.2 | 12.8 | 84.1 | 18.5 | 14.9 | **95.4** | **22.6** | 91.0 |
| | 20 | **93.0** | 11.6 | **92.9** | 18.2 | **11.9** | 93.9 | 22.7 | 89.9 |
| | 40 | 86.3 | **10.9** | 83.4 | **18.0** | 15.5 | 95.3 | 22.9 | **93.0** |
| MINE-based | 10 | 33.8 | 21.6 | 36.0 | 30.8 | 14.0 | 35.5 | 22.7 | 38.6 |
| | 20 | 41.9 | 25.0 | 49.5 | 25.4 | 22.2 | 37.5 | 33.2 | 39.0 |
| | 40 | 46.8 | 23.8 | 49.8 | 28.0 | 26.6 | 37.7 | 27.6 | 48.8 |
| Cycle loss | 10 | 55.1 | 25.4 | 64.3 | 27.8 | 17.0 | 33.1 | **22.6** | 37.4 |
| | 20 | 52.0 | 23.6 | 58.5 | 29.2 | 18.2 | 35.9 | 23.4 | 38.8 |
| | 40 | 53.8 | 20.7 | 62.8 | 22.4 | 19.3 | 31.6 | 24.5 | 35.8 |
| $\beta$-VAE | - | 24.9 | | 27.0 | | 25.6 | | 30.5 | |
| EC$^2$-VAE (c) | - | 95.2 | 11.5 | 95.1 | 18.0 | 16.8 | 57.7 | 22.6 | 57.2 |
| EC$^2$-VAE (c & s) | - | 95.2 | 13.6 | 95.1 | 19.6 | 25.2 | 96.2 | 31.0 | 91.0 |

## 4 EXPERIMENTS

We evaluate V3 on both synthetic and real data to evaluate its effectiveness and generalizability in different domains and scenarios, covering audio, image and video data. The highlight of this section is that V3 effectively learns disentangled representations of content and style, performs well on out-of-distribution generalization, and the discrete content representations manifest symbolic-level interpretability that aligns well with human knowledge. We also provide additional results in Appendix C, and the ablation study in Appendix D.

We compare V3 with three unsupervised baselines: 1) an unsupervised content-style disentanglement based on MINE (Tjandra et al., 2020a), and 2) a 2-branch autoencoder similar to our architecture choice, but trained with the cycle consistency loss after decoding and encoding shuffled combinations of $\tilde{z}^c$ and $z^s$ (Zhu et al., 2017). 3) a vanilla $\beta$-VAE (Higgins et al., 2017). Additionally, we compare with two methods with label supervision: 1) a weakly-supervised method for disentanglement named EC$^2$-VAE, in which the model is trained to predict the correct content labels from $z^c_{ij}$ as a replacement of the VQ layer, and the decoder is trained to reconstruct inputs from $z^s_{ij}$ and ground truth content labels (Yang et al., 2019; Wang et al., 2020), and 2) a fully supervised variant of EC$^2$-VAE provided with both content and style labels, in which the model learns to predict both content and style from their latent representations. We denote them as EC$^2$-VAE (c) and EC$^2$-VAE (c & s) respectively. All reported results are the average of three best-performing checkpoints on validation sets. We provide further details of model architectures in Appendix B.

### 4.1 DATASETS

**Written Phone Numbers Dataset (PhoneNums)**: We synthesize an image dataset of written digit strings on light backgrounds using 8 different ink colors, mimicking a scenario of handwritten phone numbers. The order of digits is random. All images are diversified with noises, blur, and foreground and background color jitters. Models should learn digits and colors as content and style.

**Monophonic Instrument Notes Dataset (InsNotes)**: We synthesize a dataset consisting of 16kHz monophonic music audio of 12 different instruments playing 12 different pitches in an octave. Every pitch is played for one second with a random velocity and amplitude envelope. The audio files are then normalized and processed to magnitude spectrograms. Models should learn pitches and timbres as content and style, respectively.

**Street View House Numbers (SVHN)** (Netzer et al., 2011): We select all images with more than one digit from the SVHN dataset. We crop the images to the bounding boxes of the digits and resize them to 32×48. Models should learn digits as content, their fonts, texture and colors as style. Note that the styles can be seen as from a continuous space, and the fonts in SVHN are very diverse.

Table 2: Evaluation of pitch and timbre disentanglement on InsNotes using latent retrieval. Values are reported in percentage.

| Method | $K$ | Content | | | | Style | | | |
| | | PR-AUC | | Best F1 | | PR-AUC | | Best F1 | |
| | | $z^c \uparrow$ | $z^s \downarrow$ | $z^c \uparrow$ | $z^s \downarrow$ | $z^c \downarrow$ | $z^s \uparrow$ | $z^c \downarrow$ | $z^s \uparrow$ |
|---|---|---|---|---|---|---|---|---|---|
| V3 | 12 | **89.9** | 8.9 | **90.1** | 15.1 | **9.3** | **87.5** | **15.0** | **88.0** |
| | 24 | 76.2 | 8.7 | 80.0 | 14.2 | 12.8 | 68.9 | 20.3 | 70.0 |
| | 48 | 72.2 | 8.4 | 74.4 | **14.2** | 12.3 | 72.2 | 22.0 | 71.5 |
| MINE-based | 12 | 56.4 | **7.61** | 62.0 | 14.2 | 10.3 | 61.4 | 16.9 | 63.7 |
| | 24 | 50.5 | 8.5 | 59.1 | 14.9 | 14.7 | 53.4 | 19.5 | 51.4 |
| | 48 | 44.6 | 10.2 | 54.0 | 16.5 | 13.8 | 52.1 | 18.3 | 49.7 |
| Cycle loss | 12 | 49.7 | 8.7 | 57.9 | 15.2 | 10.7 | 12.7 | 18.2 | 19.0 |
| | 24 | 47.0 | 8.7 | 54.5 | 15.2 | 14.2 | 18.9 | 19.4 | 23.1 |
| | 48 | 42.4 | 8.0 | 49.4 | 14.5 | 16.2 | 20.0 | 22.4 | 24.4 |
| $\beta$-VAE | - | 18.1 | | 20.8 | | 12.2 | | 19.0 | |
| EC$^2$-VAE (c) | - | 83.2 | 8.0 | 86.2 | 14.2 | 10.7 | 60.0 | 16.9 | 62.8 |
| EC$^2$-VAE (c & s) | - | 90.4 | 7.9 | 90.4 | 14.2 | 11.1 | 90.5 | 18.0 | 90.4 |

**Sprites with Actions Dataset (Sprites)** (ope; Yingzhen and Mandt, 2018): The Sprites dataset contains animated cartoon characters with random appearances. We use a modified version of the dataset taking video sequences of characters performing 9 different actions in random order. Models should learn the actions as content and appearances as style. Note that the styles can be seen as from a continuous space.

**Librispeech Clean 100 Hours (Libri100)** (Panayotov et al., 2015): Librispeech is a large-scale multi-speaker corpus of read English speech in various accents. We use the "clean" pool of the Librispeech dataset, where we select the 100-hour subset for training. We align the audio to 39 phonemes (24 consonants and 15 vowels) using ground truth transcriptions with the Montreal Forced Aligner (McAuliffe et al., 2017), then extract 80-dimensional log-mel spectrograms and resize them to a length of 64 frames. Models should learn phonemes as content and speakers' voice as style. The styles are considered as from a continuous space as the speakers' voices are very diverse.

## 4.2 RESULTS OF CONTENT-STYLE DISENTANGLEMENT

On PhoneNums and InsNotes where concrete style labels are available, we evaluate the models' content-style disentanglement ability by conducting a retrieval experiment to examine the nearest neighbors of every input $z^c$ and $z^s$ using ground truth content and style labels, evaluated by the area under the precision-recall curve (PR-AUC) and the best F1 score. We experiment with different codebook sizes $K$ to allow different levels of vocabulary redundancy. The results are shown in Table 1 and Table 2. We see that V3 outperforms unsupervised baselines on both datasets, and the performance is consistent across different codebook sizes $K$. V3 also outperforms EC$^2$-VAE (c) in the style retrieval task, which indicates that V3 learns better-disentangled style representations containing less content information. Visualizations of content and style latent representations learned by V3 also show clearer grouping compared to baselines, for which we refer readers to Appendix C.1.

Table 3: Linear probing accuracies (in %) for content (digit) classification on SVHN.

| Method | $K$ | $z^c \uparrow$ | $z^s \downarrow$ |
|---|---|---|---|
| V3 | 20 | **40.6** | **18.5** |
| MINE-based | 20 | 36.0 | 20.8 |
| Cycle loss | 20 | 16.8 | 21.2 |
| $\beta$-VAE | - | 21.8 | |
| Raw input | - | 21.4 | |
| EC$^2$-VAE (c) | - | 97.0 | 21.2 |

Table 4: Linear probing accuracies (in %) for content (action) classification on Sprites.

| Method | $K$ | $z^c \uparrow$ | $z^s \downarrow$ |
|---|---|---|---|
| V3 | 18 | **88.2** | **20.2** |
| MINE-based | 18 | 79.1 | 22.2 |
| Cycle loss | 18 | 86.4 | 39.7 |
| $\beta$-VAE | - | 33.2 | |
| Raw input | - | 99.0 | |
| EC$^2$-VAE (c) | - | 99.8 | 15.7 |

Table 5: Linear probing accuracies (in %) for content (phoneme) classification on Libri100.

| Method | $K$ | $z^c \uparrow$ | $z^s \downarrow$ |
|---|---|---|---|
| V3 | 80 | **52.1** | **40.4** |
| MINE-based | 80 | 28.6 | 51.6 |
| Cycle loss | 80 | 16.1 | 50.5 |
| $\beta$-VAE | - | 11.0 | |
| Raw input | - | 31.8 | |
| EC$^2$-VAE (c) | - | 78.1 | 18.2 |

Table 6: Speaker verification equal error rates (in %) with average embedding on Libri100.

| Method | $K$ | $z^c \uparrow$ | $z^s \downarrow$ |
|---|---|---|---|
| V3 | 80 | 49.5 | **42.5** |
| MINE-based | 80 | **49.9** | 45.2 |
| Cycle loss | 80 | 49.8 | 45.9 |
| $\beta$-VAE | - | 50.0 | |
| Raw input | - | 47.1 | |
| EC$^2$-VAE (c) | - | 49.2 | 49.6 |

On SVHN and Sprites where there are no style labels, we evaluate the models' disentanglement ability by linear probing on the learned representations to predict content labels. We also compare with a linear classifier on raw input features. The classifier layer is trained for one epoch before evaluated on the test set. On both datasets we allow a 100% content vocabulary redundancy, resulting in $K = 20$ for SVHN and $K = 18$ for Sprites. The resulting accuracies are shown in Table 3 and Table 4. V3 outperforms unsupervised baselines on both datasets, only trailing behind the weakly supervised EC$^2$-VAE (c) as the latter's $z^c$ space is optimized for discriminative task.

On Libri100, we evaluate the disentanglement ability by linear probing on the learned representations to predict content labels, as well as conducting a vanilla speaker verification experiment using the average embeddings of fragments in every utterance. We also allow a content vocabulary redundancy of about 100% ($K = 80$). The results are shown in Table 5 and Table 6. The content and style embeddings learned by V3 shows better performance on their respective tasks and lower performance on the other task, indicating that V3 learns better disentangled representations.

## 4.3 CONTENT CLASSIFICATION ON OUT-OF-DISTRIBUTION STYLES

We further evaluate the generalization ability of V3 on PhoneNums and InsNotes by testing the models' content classification performance on a special test set with only **unseen** styles, provided with few-shot examples. We focus on comparing V3 with the weakly supervised method EC$^2$-VAE (c) and a pure CNN classifier to evaluate the generalization ability introduced by latent disentanglement. In the $n$-shot settings, models are presented with $n$ samples of each content and new style combination. All models are continuously trained on new samples until performance stops improving. For V3, we choose the V3 versions with no codebook redundancy for comparison ($K = 10$ for PhoneNums, $K = 12$ for InsNotes) as they show a one-to-one mapping from codebook entries to content labels (see Section 4.4 and Appendix C.2 for details). We first align the learned codebook entries to ground truth content labels, and obtain classification results by the encoded content representations $z^c$. For EC$^2$-VAE, we try two different continuous training strategies: 1) using pseudo content labels from its own predictions for self-boosting, as well as training the reconstruction loss, and 2) only optimize the reconstruction loss. Additionally, we compare with EC$^2$-VAE and the CNN classifier provided with labels in continuous training. The results are shown in Table 7. Although V3 might fall behind supervised methods in the 0-shot setting, it comes to the lead in few-shot settings on both datasets as the number of extra samples increases. This indicates that V3 can learn by itself to make sense of unseen styles with only a few examples, an ability that emerges from learning representations and disentangled interpretable factors.

Table 7: Content classification accuracies (in %) on data with OOD styles.

| | Pretraining | Continuous Training | | PhoneNums | | | | InsNotes | | | |
|---|---|---|---|---|---|---|---|---|---|---|---|
| Method | Supervision | Supervision | Self-boost | 0-shot | 1-shot | 5-shot | 10-shot | 0-shot | 1-shot | 5-shot | 10-shot |
| V3 | No | No | No | 57.8 | 91.3 | **97.1** | **99.0** | 90.5 | **97.6** | **97.8** | **99.2** |
| EC$^2$-VAE (c) | Yes | No | No | **84.2** | **92.1** | 92.2 | 92.7 | 87.1 | 87.2 | 89.4 | 91.2 |
| EC$^2$-VAE (c) | Yes | No | Yes | **84.2** | 91.8 | 92.1 | 92.4 | 87.1 | 94.6 | 95.0 | 95.1 |
| CNN Classifier | Yes | No | No | 59.5 | 59.5 | 59.5 | 59.5 | **92.6** | 92.6 | 92.6 | 92.6 |
| CNN Classifier | Yes | No | Yes | 59.5 | 80.2 | 82.2 | 82.7 | **92.6** | 87.6 | 85.9 | 85.3 |
| EC$^2$-VAE (c) | Yes | Yes | No | 84.2 | 94.6 | 98.8 | 99.2 | 87.1 | 97.7 | 98.9 | 99.8 |
| CNN Classifier | Yes | Yes | No | 59.5 | 81.2 | 82.4 | 83.5 | 92.6 | 91.9 | 91.3 | 89.1 |

Table 8: Quantitative results of codebook interpretability on datasets with discrete style labels. Values are reported in percentage.

| Method | PhoneNums | | | InsNotes | | |
|---|---|---|---|---|---|---|
| | $K$ | Acc. $\uparrow$ | $\sigma \downarrow$ | $K$ | Acc. $\uparrow$ | $\sigma \downarrow$ |
| V3 | 10 | **89.2** | **0.6** | 12 | **99.8** | **0.1** |
| | 20 | **99.7** | **1.8** | 24 | **92.9** | **2.2** |
| | 40 | **99.9** | 4.5 | 48 | **90.2** | 4.5 |
| MINE-based | 10 | 40.9 | 8.1 | 12 | 13.8 | 3.8 |
| | 20 | 25.6 | 9.8 | 24 | 29.4 | 8.9 |
| | 40 | 50.6 | 6.2 | 48 | 26.9 | **3.6** |
| Cycle loss | 10 | 71.0 | 3.7 | 12 | 27.5 | 11.4 |
| | 20 | 89.6 | 4.3 | 24 | 28.5 | 11.9 |
| | 40 | **99.9** | **4.1** | 48 | 18.2 | 6.2 |

Table 9: Quantitative results of codebook interpretability on datasets without discrete style labels. Values are reported in percentage.

| Method | SVHN | | Sprites | | Libri100 | |
|---|---|---|---|---|---|---|
| | $K$ | Acc. $\uparrow$ | $K$ | Acc. $\uparrow$ | $K$ | Acc. $\uparrow$ |
| V3 | 20 | **47.6** | 18 | **98.5** | 80 | **24.7** |
| MINE-based | 20 | 26.0 | 18 | 38.3 | 80 | 10.8 |
| Cycle loss | 20 | 20.1 | 18 | 82.0 | 80 | 10.5 |

## 4.4 RESULTS OF SYMBOLIC CONTENT INTERPRETABILITY

We notice that interpretable symbols emerge in the learned codebook of V3, showing its ability of abstracting concepts from information. To evaluate the interpretability of learned content representations quantitatively, we propose two metrics: the learned content codebook accuracy and standard deviation among styles. We first align codebook entries to the ground truth content labels by their distributions on content labels, and then calculate the accuracy of codebook entries' distribution regarding their aligned labels. A well learned interpretable codebook should have entries concentrated on content labels they are aligned with, thus showing high accuracy. Also, as a good symbol is a symbol of consensus, on datasets with discrete style labels, we also quantify different styles' discrepancy of codebook entries distribution on content labels using the standard deviation ($\sigma$) of confusion matrices between codebook entries and content labels, as shown in Table 8. For datasets with no discrete style labels, we report the accuracy of codebook entries in Table 9. Visualizations of confusion matrices between the codebook and ground truth content labels can be found in Appendix C.2.

From Table 8 and Table 9, we observe that V3 shows good codebook interpretability by representing content labels with consistent codebook entries, and the consistency is kept well among styles. Table 8 also shows that V3 shows good codebook interpretability with or without vocabulary redundancy, indicating V3 does not rely on the knowledge of the number of content classes to learn interpretable symbols.

Qualitatively, we perform content and style recombination by traversing all content codebook entries and decoding them with a fixed style representation. If the learned codebook has good interpretability, the decoding results of recombined content and style representations should show meaningful content changes, and retain consistent styles. Here, we focus on comparing the recombination results of V3 and baselines on SVHN, where we first encode $z^{\mathrm{s}}$ from example fragments, and then recombine it with all $K$ content codebook entries. More experiment results on PhoneNums and InsNotes can be found in Appendix C.2 and our demo website.

From Figure 3, we observe that V3 generates images with clear content changes and consistent styles when recombining content and style representations. Although images generated by V3 may not cover all possible contents as many of them never appear in the training set in the given styles, the content are almost all recognizable digits and V3 can even "imagine" reasonably what a digit would look like in a new font and color. In contrast, the baselines generate images with either mixed content and style information, or very subtle changes in content that are hard to interpret. This comparison

Figure 3: Comparison of generated images by recombining $z^s$ from given sources in SVHN and all $z^c$ in the learned codebook.

not only validates the interpretability of V3's learned codebook resulted from successful content-style disentanglement, but also demonstrates the potential of V3 in style transfer and content editing tasks.

## 5 LIMITATION

We have identified several limitations in our V3 method that necessitate further investigation. First, while V3 achieves good disentanglement and symbolic interpretability, it is not flawless — samples of different contents (say images of "8" and "9") may be projected into the same latent code. Inspired by human learning, which effectively integrates both mode-1 and mode-2 cognitive processes, we aim to enhance V3 by incorporating certain feedback or reinforcement. This adaptation could also facilitate the application of V3 to more complex domains such as general image or video. Also, V3 focuses on learning the atomic content symbols that compose meanings of data samples, but the current approach does not learn the meanings of individual symbols or those emerging from specific symbol permutations. Additionally, V3 is currently optimized to disentangle content and style from data samples that include defined fragments. Extending this capability to unsegmented data of large vocabularies, such as continuous audio, represents a significant area for future development. Furthermore, V3 assumes that content elements do not overlap, which does not hold in cases of polyphonic music or mixed audio. Addressing this challenge will require a more sophisticated approach that considers the hierarchical nature of content.

## 6 CONCLUSION

In conclusion, we contributed an unsupervised content-style disentanglement method named V3. V3's inductive bias is domain-general, intuitive, and concise, solely based on the meta-level insight of the statistical difference between content and style, i.e., their distinct variance-invariance patterns reflected both within and across data samples. Experiment results showed that V3 not only outperforms the baselines in terms of content-style disentanglement, but also demonstrates superior generalizability on OOD styles compared to supervised methods, and achieves high interpretability of learned content symbols. The effectiveness of V3 generalizes across different domains, including audio, image, and video. We believe that V3 has the potential to be applied to emergent knowledge in general, and we plan to extend our method to more complex tasks and domains in the future.

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

APPENDICES

The appendix is structured into 5 main parts. Appendix A provides specifics about the datasets involved in the paper. Appendix B presents implementation and training details of V3 and baseline methods. Appendix C provides additional experiment results and especially visualizations for better understanding. Appendix D presents an ablation study on the V3 model. Finally, we provide an analysis on learning content and style in Appendix E.

# A DATASET DETAILS

## A.1 PHONENUMS

The written phone numbers dataset is designed to represent a clear content and style separation to human. We use the Kristen ITC font for the style because its digits look similar to handwritten digits and are easy to distinguish. We render the digits from 0 to 9 on a light background of RGB (10, 10, 10) using the foreground colors listed in Table 10. For more randomness, we first jitter the foreground

Table 10: List of colors and their corresponding RGB values. Colors only used for out-of-distribution experiments in Section 4.3 are marked with *.

| RGB Values # | Color |
|---|---|
| (8, 8, 8) | Black |
| (8, 8, 248) | Blue |
| (8, 128, 8) | Green |
| (248, 8, 8) | Red |
| (8, 128, 128) | Teal |
| (128, 8, 128) | Purple |
| (248, 163, 8) | Orange |
| (163, 47, 47) | Brown |
| (248, 188, 199) | Pink * |
| (243, 128, 115) | Salmon * |
| (248, 210, 8) | Gold * |
| (8, 248, 8) | Lime * |
| (8, 248, 248) | Cyan * |
| (248, 8, 248) | Magenta * |
| (128, 128, 128) | Gray * |
| (200, 133, 67) | Peru * |

and background colors by a noise from -2 to 2 along every channel, then add a small Gaussian noise. We then translate all digits vertically or horizontally by a random number of pixels between -2 and 2. Lastly, we add a random Gaussian blur effect. Our dataset contains 100000 images in total, each of which has 10 digits. The dataset is split into the train set, validation set, and test set with a ratio of 8:1:1. Some samples of the dataset can be viewed in Figure 4

Figure 4: Left: example training data in PhoneNums. Right: example data for out-of-distribution evaluation in PhoneNums.

## A.2 INSNOTES

In InsNotes, we collect monophonic music audio played by different instruments. This dataset is also designed based on the understanding that this domain exhibits content and style concepts that are

clear to human - music pitches and instrument timbres. The dataset consists of monophonic music audio rendered from 12 instruments playing 12 different pitches in an octave, which corresponds to MIDI numbers from 60 to 71. All the instruments selected have little exponential decays, so their timbres can be well represented with short audio samples. In the out-of-distribution generalization experiment in Section 4.3, we select four unseen instruments: two with slight exponential decays and two with strong attacks and distinct exponential decays, making the task particularly challenging. We list the instruments involved as well as the specific MIDI program selected in Table 11.

Table 11: List of instruments and their corresponding MIDI program numbers. Instruments only used for out-of-distribution generalization are marked with *.

| Program # | Instrument |
|---|---|
| 19 | Pipe organ |
| 21 | Accordion |
| 22 | Harmonica |
| 41 | Viola |
| 52 | Choir aahs |
| 56 | Trumpet |
| 59 | Muted trumpet |
| 64 | Soprano sax |
| 68 | Oboe |
| 71 | Clarinet |
| 72 | Piccolo |
| 75 | Pan flute |
| 0 | Grand piano * |
| 4 | Tine electric piano * |
| 73 | Flute * |
| 78 | Irish flute * |

For every instrument, we play every pitch for one second one by one with a random velocity between 80 and 120 until every pitch is played 10 times. We synthesize 100 such takes at 16kHz using a soundfont library for each instrument and further diversify every note by adding a random amplitude envelope to each note. The added amplitude envelope is either a linear curve or a sinusoidal curve, starting and ending at a random amplitude factor between 0.8 and 1.2. The audio files are then normalized and processed with short-time Fourier transform (STFT) with the FFT size of 1024 and hop size of 512 to obtain the magnitude spectrograms, which results in a $512 \times 32$ matrix for each note. To avoid possible overlap between adjacent notes, we add a 0.056 second pause in between, resulting in one transition frame in the spectrogram. Our dataset contains 1200 audio files in total, each of which has 120 notes. The dataset is split into the train set, validation set, and test set with a ratio of 8:1:1.

## A.3   SVHN

The Street View House Numbers (SVHN) dataset is a real-world dataset that contains images of house numbers collected from Google Street View, and digit-level bounding boxes (Netzer et al., 2011). Examples of the dataset can be viewed in Figure 5. The dataset originally consists of 73257 digits for training, 26032 digits for testing, and 531131 additional, somewhat less difficult samples, as the extra partition. We split the extra partition into additional training, validation and testing sets with a ratio of 8:1:1. For the content-style disentanglement task, we select all images with at least two labeled digits, and resize the bounding boxes to $32 \times 48$ pixels. The dataset is preprocessed by normalizing the pixel values to the range of [0, 1]. Compared to PhoneNums, although both being image datasets with digits as content, SVHN is significantly more challenging for the following reasons: 1) The digits in SVHN can be very blurry compared to that in PhoneNums; 2) The digits in SVHN come with more flexible styles in a totally continuous space, involving different fonts, thicknesses, inclinations, colors, and so on; 3) In every SVHN image, the style variation among digits can be more significant than that in PhoneNums, as there can be environmental factors like shadows; 4) The bounding boxes are not always tight, clean and complete, and the digits are not always centered in the image; 6) The classes are very imbalanced. Almost all images come with an 0, 1 or 2 but very few of them have 8

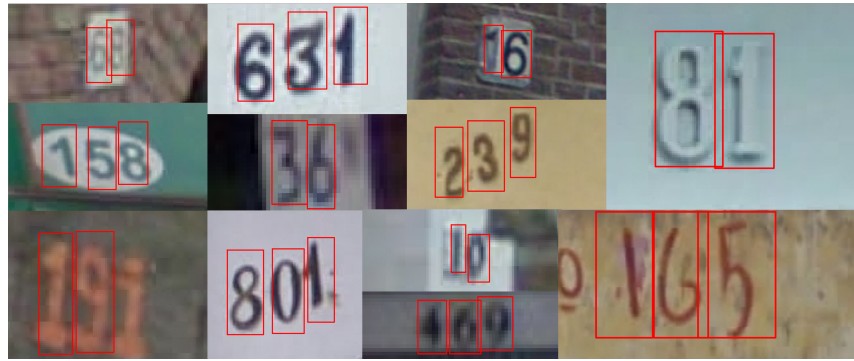

Figure 5: Example data in the original SVHN dataset. The digits in the images are bounded by the red boxes.

or 9; 5) Most importantly, house numbers are generally very short strings. Among images with at least two digits, 57.6% of them have exactly two digits, which means for most styles, there is not a full coverage of all digits, and during training, V3 only has a highly incomplete view of the full content vocabulary. We choose SVHN to demonstrate the robustness of V3 in learning content and style disentanglement in a much more challenging setting.

### A.4 SPRITES

The original Sprites dataset, collected from ope and adopted by Yingzhen and Mandt (2018), contains animated cartoon characters in the pixel graphic style with random appearances and actions. The original Sprites dataset contains animations of six different actions in four perspectives. We collect the Sprites with Actions dataset used in this study by selecting 3 distinct actions in 3 perspectives, resulting in 9 different actions in total, and rendering videos of characters performing actions from these 9 categories randomly, using the critical frames from each action animation. The dataset contains 2160 videos in total, each of which has 9 frames. The characters differ in their hair, body, top and bottom, forming 2160 unique characters in total. We use 80% of the characters for training and the rest for validation and testing. Examples of the dataset can be viewed in the right of Figure 1.

### A.5 LIBRI100

The Libri100 dataset is a subset of the LibriSpeech dataset (Panayotov et al., 2015), containing the "train-clean-100", "dev-clean", and "test-clean" subsets. There are 331 different speakers in total, in which 165 are female and 166 are male. There are no overlapping speakers between the train, validation, and test divisions. Given audio files and ground truth transcriptions, we align the audio with the 39 phonemes used in English using the Montreal Forced Aligner (McAuliffe et al., 2017). After normalizing the cropped fragments, we extract the mel spectrograms with a window size of 16ms, hop size of 5ms, and 80 mel bands. The 39 phonemes are indexed as shown in Table 12.

## B IMPLEMENTATION DETAILS

### B.1 MODEL ARCHITECTURE

On InsNotes, we instantiate V3 model using a ResNet18 encoder and a ResNet18T decoder with bottlenecks (He et al., 2016). In the encoder, the number of channels in the first convolutional layer is set to 64, and gradually increase to 512 in the last layer. The first half of the encoder uses a kernel size of 9 and the second half uses a kernel size of 5. The decoder is symmetric to the encoder. The latent dimension is set to 512. The total number of trainable parameters is 55M.

On PhoneNums and Sprites, we instantiate V3 model using a ResNet encoder and a ResNetT decoder half deep as the pitch and timbre learning task. Similarly, the number of channels in the first convolutional layer is set to 16, and gradually increase to 256 in the last layer. The encoder uses a

Table 12: List of phonemes with their indices.

| Index | Phoneme | Index | Phoneme | Index | Phoneme |
|-------|---------|-------|---------|-------|---------|
| 0 | eh | 13 | ao | 26 | l |
| 1 | z | 14 | ey | 27 | k |
| 2 | s | 15 | hh | 28 | m |
| 3 | uw | 16 | y | 29 | ch |
| 4 | aw | 17 | f | 30 | ng |
| 5 | oy | 18 | r | 31 | t |
| 6 | dx | 19 | g | 32 | w |
| 7 | dh | 20 | v | 33 | ae |
| 8 | uh | 21 | ah | 34 | iy |
| 9 | aa | 22 | er | 35 | th |
| 10 | d | 23 | ow | 36 | ay |
| 11 | p | 24 | sh | 37 | ih |
| 12 | n | 25 | b | 38 | jh |

kernel size of 5. The decoder is symmetric to the encoder. The latent dimension is set to 512. The total number of trainable parameters is 20M on PhoneNums and 25M on Sprites.

On SVHN, we add one more ResNet layer in every ResBlock on top of the ResNet encoder used in PhoneNums and Sprites. The number of channels in the first convolutional layer is set to 32, and gradually increase to 512 in the last layer. The first half of the encoder uses a kernel size of 5 and the second half uses a kernel size of 3. The decoder is symmetric to the encoder. The latent dimension is set to 768. The total number of trainable parameters is 37M.

On Libri100, we use a similar architecture as InsNotes, but with a maximum number of channels of 256. We deepen the encoder with 2 more ResNet blocks in each layer. The total number of trainable parameters is 24M.

We use the same neural network architecture as V3 for the MINE-based baseline and the cycle loss-based baseline, except that the style branch of the MINE-based method has a variational latent layer. For the MINE-based baseline, we use a 3-layer multi-layer perceptron with 512 hidden units to estimate the mutual information. For the supervised baselines $EC^2$-VAE (c), we replace the VQ layer of the content branch with a linear layer projecting to the dimension of prediction logits. Besides, the encoder output of the style branch are mean and log variance vectors instead of representation vectors, which means the style branch is a variational autoencoder (VAE) (Kingma and Welling, 2013). For the fully supervised baseline $EC^2$-VAE (c/s), we project the reparameterized style vectors to the dimension of prediction logits.

### B.2 TRAINING DETAILS

For all models, we use the Adam optimizer with a learning rate of 0.001 (Kingma and Ba, 2014). The fragment sizes on PhoneNums, InsNotes, SVHN and Sprites are set to 10, 12, 2 and 6, respectively. The relativity $r$ is set to 15, 15, 5, 10 and 5 on PhoneNums, InsNotes, SVHN, Sprites and Libri100, respectively (Generally, we recommend setting a higher $r$, such as 15, on datasets with clean content and style separation, and setting a lower $r$, such as 5, on more complex datasets where reconstruction might need to be emphasized more.). The V3 loss weight $\beta$ is defaultly set to 1 on InsNotes task, and 0.1 in other datasets. For all VQ-based models, we update the codebooks using exponential moving average with a decay rate of 0.95 (Van Den Oord et al., 2017). The commitment loss weight $\alpha$ is set to 0.01. On PhoneNums and InsNotes, we set a threshold of $\frac{n}{10K}$ for dead code relaunching to improve codebook utilization, where $n$ is total the number of fragments in a batch. On SVHN, as most images have only 2 or 3 digits, we concatenate fragments in different samples for a higher content coverage in practice to stabilize training. Similarly, on Librispeech, as many consonant phonemes do not exhibit distinct styles like vowels, we also smooth the styles by taking the average of adjacent fragments in practice. For MINE-based baseline models, we update the MINE network once every global iteration using the Adam optimizer and adaptive gradient scaling Tjandra et al. (2020a); Belghazi et al. (2018). The learning rate of the MINE network is set to 0.0002.

We train all models using an exponential decay learning rate scheduler, and take the model with the best validation loss as the final model. All models are trained on a single Nvidia RTX 4090 GPU. The V3 loss should decay to zero within a few epochs after training starts. All supervised learning methods converge within 2 hours, while the converging time of all unsupervised learning methods differs from 5 hours to 24 hours.

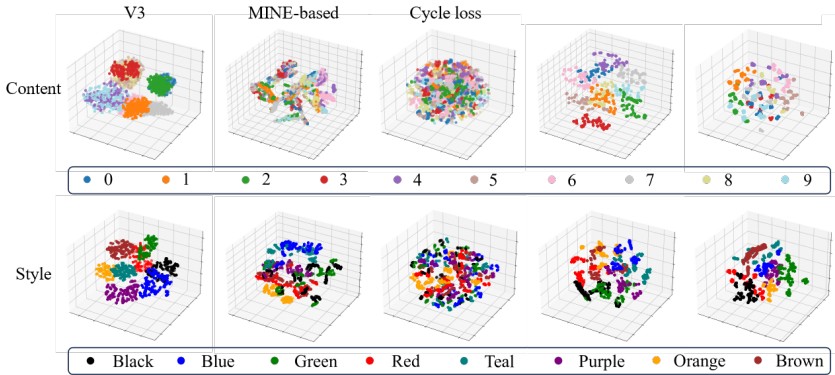

Figure 6: t-SNE visualization of the learned digit (content) and color (style) representations on PhoneNums when there is no codebook redundancy ($K = 10$).

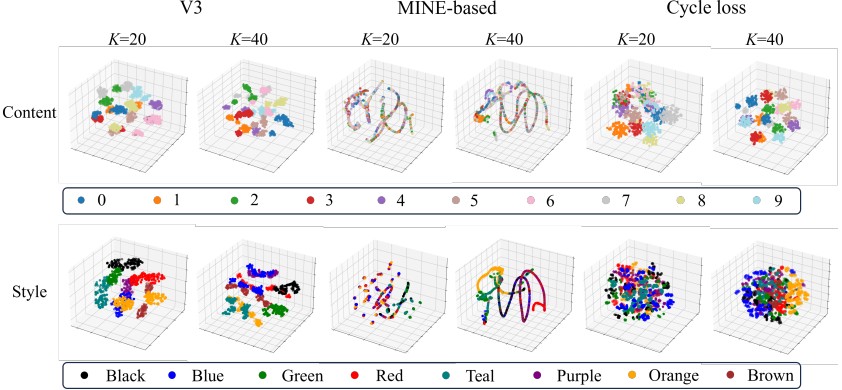

Figure 7: t-SNE visualization of the learned digit (content) and color (style) representations on PhoneNums when the codebooks are redundant.

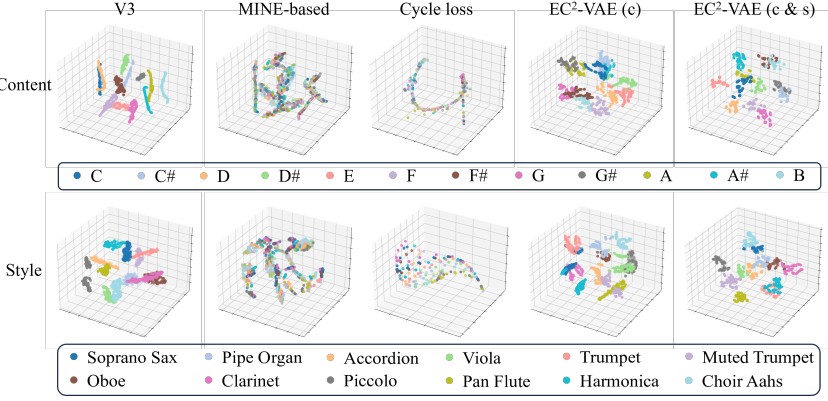

Figure 8: t-SNE visualization of the learned pitch (content) and timbre (style) representations on InsNotes when there is no codebook redundancy ($K = 12$).

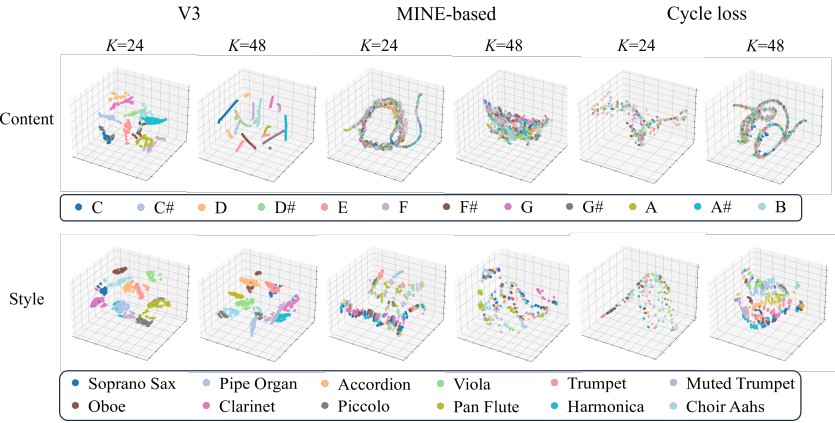

Figure 9: t-SNE visualization of the learned pitch (content) and timbre (style) representations on InsNotes when the codebooks are redundant.

## C   MORE EXPERIMENT RESULTS

In this part, we provide extra experiment results in addition to the results in Section 4. We will focus more on the visualizations of the learned content and style representations, and the alignment between the learned codebooks and the ground truth content labels.

### C.1   RESULTS OF CONTENT-STYLE DISENTANGLEMENT

This section provides 3-dimensional t-SNE visualization results of the learning content and style representations in support of Section 4.2. We show that on different datasets and under different $K$ settings, content and style representations learned by V3 show the clearest groupings compared to baselines, and the groupings match well with ground truth content and style labels.

On PhoneNums, we first visualize with t-SNE the learned content and style representations when there is no codebook redundancy ($K = 10$), and color them by the ground truth content or style labels. We set $K = 12$ for learning digits and colors. The results are shown in Figure 6. We can see that V3 learns clearer content and style representations in groups compared to unsupervised baselines. When the codebooks contain redundancy, the results are shown in Figure 7. We can see that V3 still achieves the clearest content and style grouping.

On InsNotes, the visualizations of $z^c$ and $z^s$ when $K = 12$ are shown in Figure 8, and the visualizations when codebook is redundant are shown in Figure 9. Both results also show V3 groups content and style better than baselines.

On SVHN, the visualizations of $z^c$ are shown in Figure 10. Since SVHN does not have discrete style labels, we only show the grouping of content representations. V3 is trained at $K = 20$. Although not as good as the results shown in Figure 6 on PhoneNums, V3 still achieves the best grouping of digits with learned content representations among unsupervised methods.

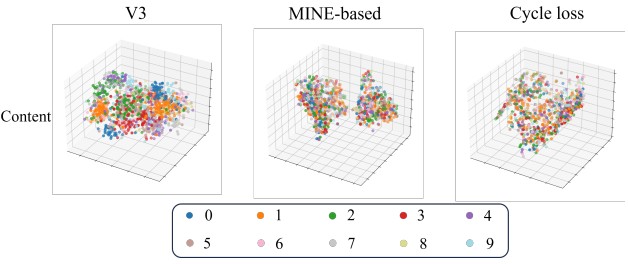

Figure 10: t-SNE visualization of the learned content (digit) representations on SVHN.

On Sprites, there is no discrete style label either. Figure 11 shows the t-SNE visualizations of $z^c$, and V3 is trained at $K = 18$. Both V3 and Cycle loss achieve good content grouping, but it is observable that some clusters of the cycle loss $z^c$ have broken into several subclusters, indicating that there is still content and style entanglement. This is also supported by Section 4.2.

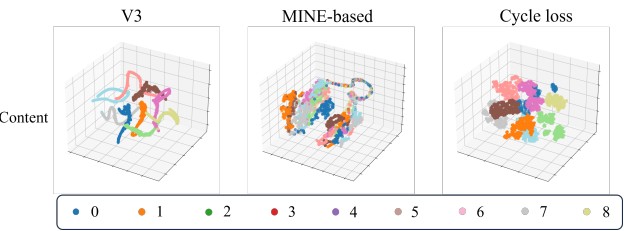

Figure 11: t-SNE visualization of the learned content (action) representations on Sprites. The 10 colors refer to 10 different actions.

## C.2 RESULTS OF SYMBOLIC CONTENT INTERPRETABILITY

This section provides intuitive visualizations about how the learned content codebook entries align with ground truth content labels in Section 4.4. We first collect frequencies of every content encoded to every codebook entry, and then permute the codebook to make the confusion matrix look like an eye for a clear alignment. Then we plot heatmaps of confusion matrices between codebook entries (vertical axes) and content labels (horizontal axes).

On PhoneNums and InsNotes, we plot the confusion matrices under different $K$ settings in Figure 12 and Figure 13, respectively. The results show V3 achieves the clearest symbol interpretability in all $K$ settings. Results on SVHN and Sprites are shown in Figure 14 and Figure 15. On Sprites, both V3 and cycle loss learns codebooks with good interpretability, but V3 still has fewer misclassifications. Although V3 does not learn a clear on-to-one codebook entry to content label mapping on SVHN, it still shows a clearer alignment relationship than other methods. An interesting fact is the order of learning we observe during training — V3 usually first distinguish 0 and 1, then start to understand 2 is different, then 3. It often confuses between 5 and 6 and between 1 and 7, and it usually fails to learn 8 and 9. This human-like learning trajectory might be subject to both the ratio of content classes and their pairwise similarities in shape. The similar phenomenon is observed in the confusion matrices on Libri100 as shown in Figure 16. V3 distinguishes between consonants and vowels pretty well, and confuses between "z" and "s", and between "n" and "ng", which are phonetically similar.

To further investigate the disentanglement ability of models, we perform latent representation recombination using the trained models. Figure 3 has already demonstrated the results on SVHN of decoding a fixed $z^s$ with every $z^c$. Here we show the results on PhoneNums, where instead of using a fixed $z^s$ encoded from an example, we compute the mean $z^s$ of all fragments from a class as its style representations for decoding. We select the V3 model with $K = 10$, align codebook entries with digit labels, and enumerate all combinations of $z^c$ and $z^s$. We present the results in Figure 17. Compared to baselines, V3 can fairly well reconstruct the involved 8 colors and the digits from 0 to 9, even though it is not informed with any discrete labels during training. In contrast, the MINE-based baseline and the cycle loss baseline fail to distinguish the digits, although the color reconstruction is not bad. They generate blurry digits that look like "5", "8" or "6", which are the most conservative choices. As for results on the music dataset InsNotes, we refer you to our web demo page for an interactive experience.

## D ABLATION STUDY

For ablation, we experiment with another type of variability measurement $\nu_k(\cdot)$, which is standard deviation (SD). Besides, we train four variants of V3, each without one of the four regularization terms defined in Equation 9-12. We conduct experiments on PhoneNums and InsNotes, two datasets with style labels available, and evaluate the content and style disentanglement performances. The results are reported in Table 13 and Table 14. It can be seen that $\nu_k = \mathrm{SD}$ does not work as well as $\nu_k = \mathrm{MPD}$, which can be explained by its weakness in constraining multi-peak content distributions

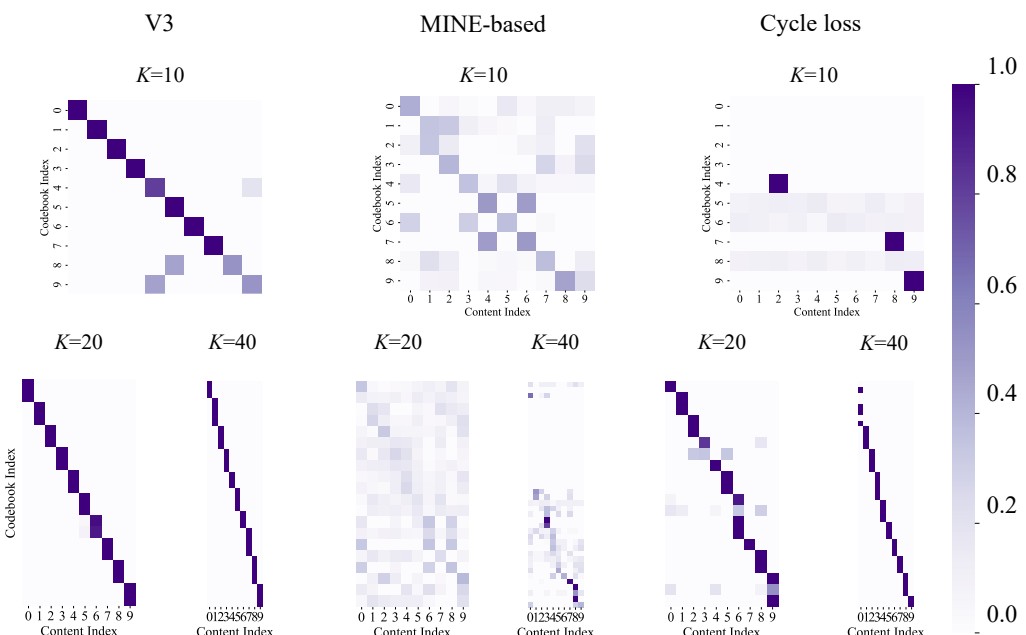

Figure 12: Confusion matrices of learned codebooks on PhoneNums. The horizontal axes show digit labels from "0" to "9", and the vertical axes show codebook atoms sorted by ground truth digit labels.

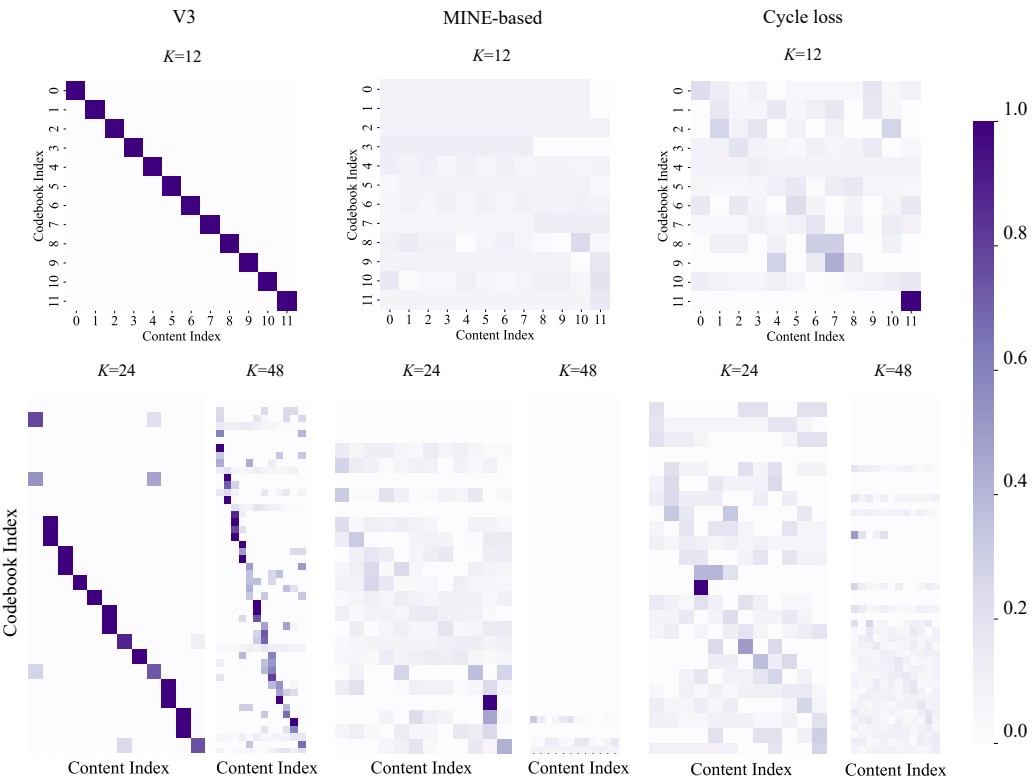

Figure 13: Confusion matrices of learned codebooks on InsNotes. The horizontal axes show pitch labels from "C" to "B", and the vertical axes show codebook atoms sorted by ground truth pitch labels.

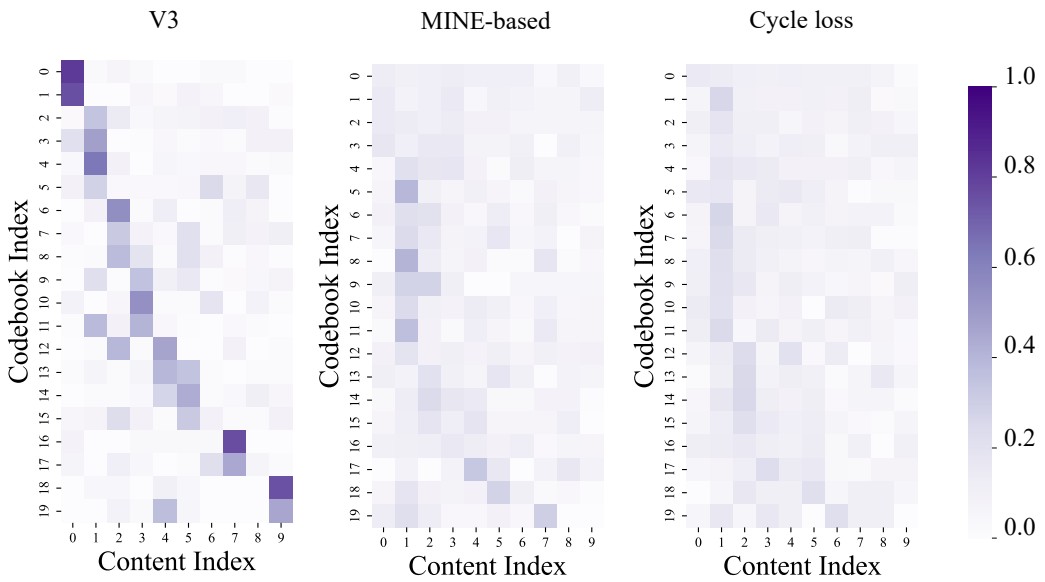

Figure 14: Confusion matrices of learned codebooks on SVHN. The horizontal axes show digit labels from "0" to "9", and the vertical axes show codebook atoms sorted by ground truth digit labels.

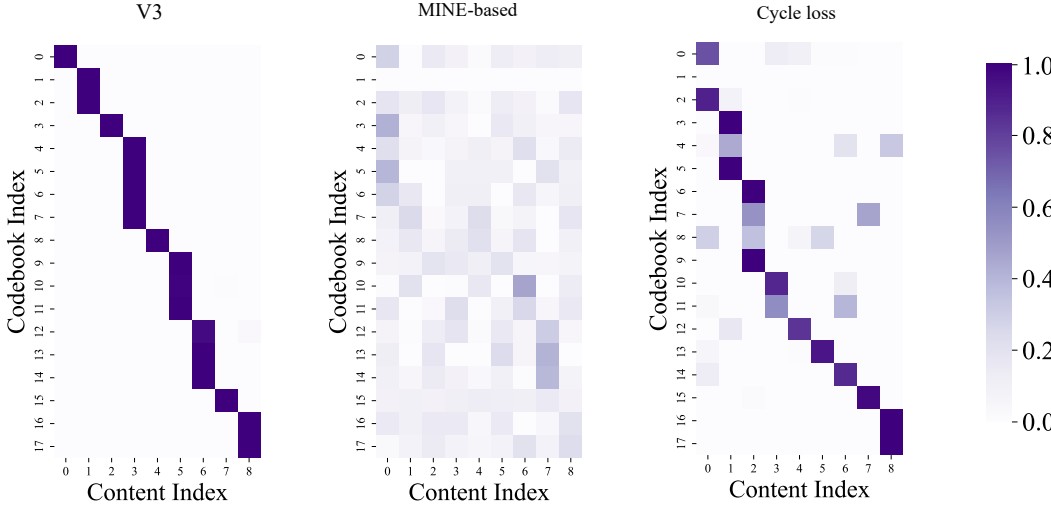

Figure 15: Confusion matrices of learned codebooks on Sprites. The horizontal axes are different action labels, and the vertical axes show codebook atoms sorted by ground truth action labels.

within samples. It is also worth noting that V3 sometimes performs fairly well even when discarding one of its terms. In these cases, we observe a decrease in the discarded loss even if we do not explicitly optimize for it. We suspect this is due to the robustness of V3 constraints as reflected in the symmetric relationships among the four losses—we can enforce three relations, and the fourth one may fall into the right place automatically. However, in practice, it is difficult to tell the one term to free beforehand as it is also related to detailed content and style variations in specific domains. As a result, the V3 constraints as a whole shows robust performance across domains.

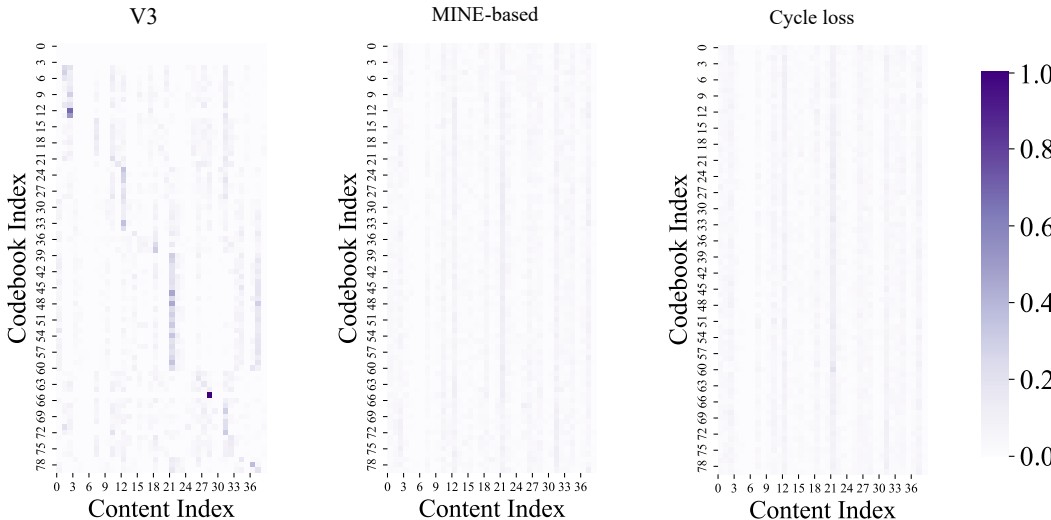

Figure 16: Confusion matrices of learned codebooks on Libri100. The horizontal axes are different phoneme labels, and the vertical axes show codebook atoms sorted by ground truth phoneme labels.

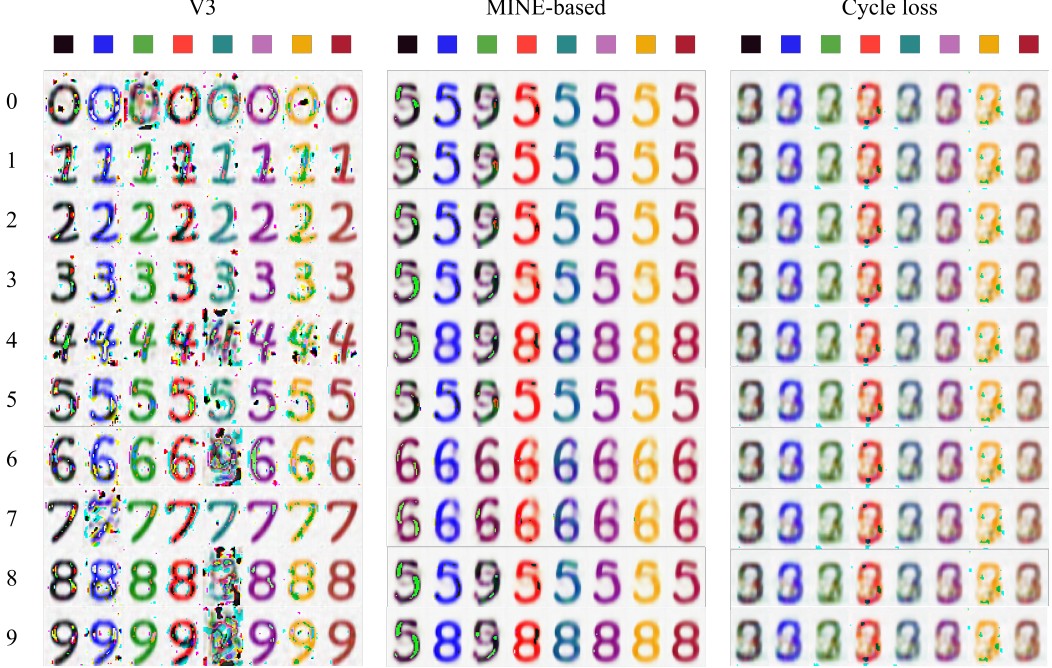

Figure 17: Comparison of generated digits by recombining content and style latents using unsupervised methods trained on PhoneNums.

# E DISCUSSION

**Connection between Content-Style Disentanglement and OOD Generalizability:** Disentanglement can intuitively boost OOD generalization for several key reasons. By separating different factors, like content and style, the model can focus on the important features without getting distracted by irrelevant variations. This separation makes the model more robust to changes. For instance, if the style changes in an OOD sample while the content remains similar, the model might still recognize and process the content effectively. Additionally, disentangled representations often lead to more

Table 13: Ablation study of V3 settings on content-style disentanglement performance on PhoneNums. Values are reported in percentage.

| Method | $K$ | Content | | | | Style | | | |
| | | PR-AUC | | Best F1 | | PR-AUC | | Best F1 | |
| | | $z^{\mathrm{c}} \uparrow$ | $z^{\mathrm{s}} \downarrow$ | $z^{\mathrm{c}} \uparrow$ | $z^{\mathrm{s}} \downarrow$ | $z^{\mathrm{c}} \downarrow$ | $z^{\mathrm{s}} \uparrow$ | $z^{\mathrm{c}} \downarrow$ | $z^{\mathrm{s}} \uparrow$ |
|---|---|---|---|---|---|---|---|---|---|
| V3 | 10 | 83.2 | 12.8 | 84.1 | 18.5 | **14.9** | **95.4** | 22.6 | **91.0** |
| V3 ($\nu_k = \mathrm{SD}$) | 10 | 42.7 | 17.9 | 53.5 | 21.5 | 18.2 | 49.9 | 24.7 | 51.6 |
| V3 (w/o $\mathcal{L}_{\mathrm{content}}$) | 10 | 43.8 | 13.8 | 51.2 | 18.9 | 18.1 | 90.8 | **22.4** | 87.5 |
| V3 (w/o $\mathcal{L}_{\mathrm{style}}$) | 10 | 64.6 | 13.2 | 70.3 | 18.7 | 17.7 | 87.8 | 24.5 | 83.9 |
| V3 (w/o $\mathcal{L}_{\mathrm{fragment}}$) | 10 | **96.3** | **11.7** | **98.9** | **17.9** | 15.9 | 90.1 | 23.2 | 88.5 |
| V3 (w/o $\mathcal{L}_{\mathrm{sample}}$) | 10 | 47.0 | 12.7 | 57.8 | 18.9 | 15.4 | 88.4 | 24.7 | 85.3 |

Table 14: Ablation study of V3 settings on content-style disentanglement performance on InsNotes. Values are reported in percentage.

| Method | $K$ | Content | | | | Style | | | |
| | | PR-AUC | | Best F1 | | PR-AUC | | Best F1 | |
| | | $z^{\mathrm{c}} \uparrow$ | $vz^{\mathrm{s}} \downarrow$ | $z^{\mathrm{c}} \uparrow$ | $z^{\mathrm{s}} \downarrow$ | $z^{\mathrm{c}} \downarrow$ | $z^{\mathrm{s}} \uparrow$ | $z^{\mathrm{c}} \downarrow$ | $z^{\mathrm{s}} \uparrow$ |
|---|---|---|---|---|---|---|---|---|---|
| V3 | 12 | **89.9** | 8.9 | **90.1** | 15.1 | **9.3** | **87.5** | **15.0** | **88.0** |
| V3 ($\nu_k = \mathrm{SD}$) | 12 | 12.9 | 9.9 | 17.5 | 15.0 | 16.3 | 24.7 | 24.1 | 36.5 |
| V3 (w/o $\mathcal{L}_{\mathrm{content}}$) | 12 | 19.2 | 9.3 | 28.0 | 14.3 | 13.6 | 66.2 | 19.0 | 68.4 |
| V3 (w/o $\mathcal{L}_{\mathrm{style}}$) | 12 | 72.1 | 8.9 | 14.2 | 84.0 | 13.7 | 78.7 | 23.6 | 79.0 |
| V3 (w/o $\mathcal{L}_{\mathrm{fragment}}$) | 12 | 26.0 | 12.1 | 35.7 | 17.6 | 13.5 | 53.7 | 20.1 | 56.7 |
| V3 (w/o $\mathcal{L}_{\mathrm{sample}}$) | 12 | 86.4 | **7.9** | 89.3 | **14.2** | 11.3 | 50.7 | 19.4 | 56.2 |

generalized features, enabling the model to identify important patterns that are invariant across different distributions. This property facilitates easier transfer learning because models with disentangled representations can be more readily fine-tuned for new tasks, as supported by our experiments in Section 4.3.

**Connection between Content-Style Disentanglement and Symbolic Interpretability:** In Section 4.2 and Section 4.4, we separately examined content-style disentanglement and symbolic-level interpretability. This discussion now seeks to understand how these elements are interconnected—specifically, whether V3's disentangled representation space inherently improves symbolic-level interpretability.

The transition from purely observational data to symbolic representation remains an open question in cognitive science and artificial intelligence. We suggest that robust content-style disentanglement is closely linked to better symbolic interpretability, as evidenced in Tables 1 to 4, 8 and 9. These figures show that both V3 and supervised methods, which achieve better disentanglement, also provide superior interpretability compared to methods less effective in disentanglement (supervised classification — while it is not — can here be viewed as another form of interpretable VQ symbols). Additionally, as illustrated in Figures 6 to 11 , well-disentangled style spaces (meaning they contain less content information) see well-formed clusters, which can facilitate straightforward postprocessing for discrete and symbolic labeling.

**Connection between Content-Style Disentanglement and Philosophy:** The statistical patterns and mutual relationship between content and style closely correspond to the philosophical concept of Yin and Yang, a fundamental duality in universal balance and dynamics. In the famous Yin-Yang diagram, Yin and Yang are equally divided and composed of two identical shapes (the fish-like swirl and the small dot) with opposing colors (light and dark), together forming the completeness of the world Wilhelm et al. (2001).

We can interpret the two fundamental shapes as the two axes along which data is observed — the swirl represents observation across samples, while the dot represents observation across fragments within samples. The two colors signify two kinds of dynamics — light denotes variant, and dark denotes invariant. As a result, the duality of Yin and Yang becomes the duality of content and style. Content shows variability within samples (the light dot) and invariability of vocabulary across samples (the

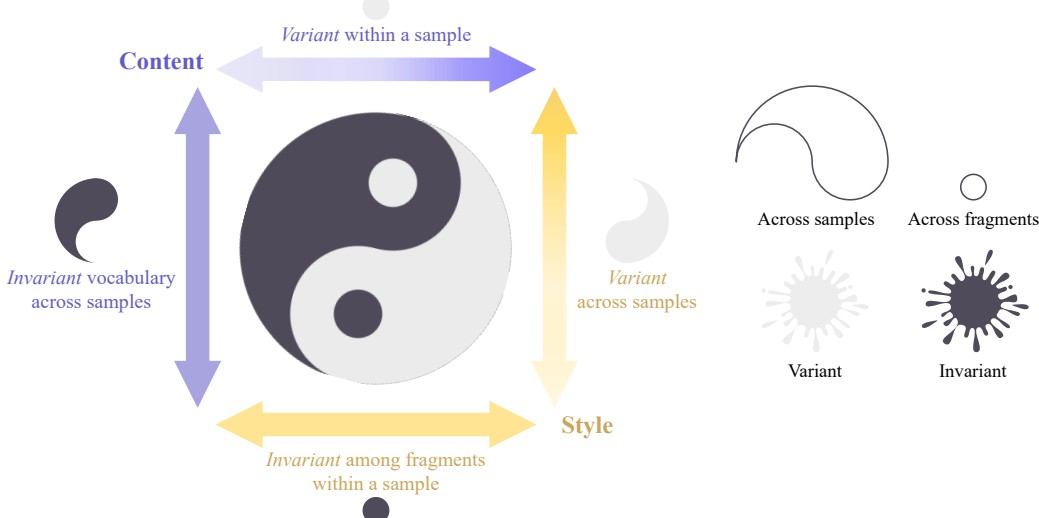

Figure 18: An illustration of the correspondence between the content-style duality and the Yin-Yang duality.

dark swirl), while style shows variability across samples (the light swirl) and invariability within a sample (the dark dot). They can be disentangled from data as two components, yet neither can exist alone.

**V3 and related works:** We explicate the difference and connection between V3 and several other most relevant works as below.

- InfoGAN (Chen et al., 2016): InfoGAN is similar to our approach in that both models learn interpretable representations and decouple these representations from the data. However, there are several key differences: 1) Each representation in InfoGAN is of very low dimensionality; 3) The specific aspects learned are less controllable, while V3 focuses on learning the distinctions of content and style; 3) GAN is known to be less unstable in training than autoencoders and VAEs, and it is a framework more for generative modeling than representation learning.

- DSAE and variants (Hsu et al., 2017; Yingzhen and Mandt, 2018; Bai et al., 2021; Luo et al., 2022; 2024): DSAE shares similar insights with V3 regarding the intrinsic relationship between content and style, but it primarily focuses on the invariability of style and the variability of content within a sample, giving less attention to the invariability of content vocabulary and the variability of style across a broader scope. Other distinctions include: 1) DSAE focuses on learning a fixed style representation for a whole sample, which may struggle with samples where the style varies, such as singing performances that feature both chest voice and falsetto, or instrument performances with multiple articulations; 2) The DSAE family requires access to the entire sequence when encoding style; 3) Most importantly, the content learned in DSAE is context-dependent, while V3 emphasizes on learning more universal content representations.

- VICReg (Bardes et al., 2022): V3 has a similar form of loss function as VICReg, and both models leverage variance and invariance among entities to help training representation learning frameworks. However, VICReg focuses on learning a single representation for each entity, while V3 focuses on learning disentangled representations. Also, VICReg learns discriminative representations without clear interpretability, but V3 learns interpretable content symbols, and has a decoding ability to recombine content-style pairs. In fact, we draw on their mathematical representations and the idea of using regularization to prevent latent representation from collapsing, a concept also advocated by LeCun (2022).

