# OpenReview forum: "Unsupervised Disentanglement of Content and Style via Variance-Invariance Constraints"
_ICLR.cc/2025/Conference — ICLR 2025 Poster_

### Official Review · Reviewer_BaC3 · 2024-11-01

**Soundness:** 3
**Presentation:** 4
**Contribution:** 3
**Rating:** 8
**Confidence:** 5

**Summary:**

The authors propose a method to learn representations from an adapted VQ-VAE that disentangle content and style based on their different statistical characteristics: style is more constant over patches of a sample, whereas content can vary.

**Strengths:**

The paper is well written and easy to follow. The method is well motivated and described and results are nicely presented.

**Weaknesses:**

I have few comments on the main body of the paper, so please do not see this as a "few line review", I just do not have much to say.

**Experiments**:
* 3 of 4 are on synthetic data with only SVHN being a "natural" data set and even that is cropped to the task.
* It is of course right to demonstrate that the model performs well on data that follows its assumptions.
* However, it would be informative (a) to see results on more natural datasets, and (b) to see how the model performs where assumptions do not hold so well, if only as a benchmark for future improvement.

**Questions:**

none

---

> ### Author Response · Authors · 2024-11-25
>
> We are grateful for your acknowledgment of our work and your constructive feedback!
> Following your suggestions, we have tested V3 on Librispeech, a large-scale real-world speech dataset. Despite the complexity of the new problem, V3 still shows better disentanglement ability and good content codebook interpretability compared to baselines, proving the universality of V3. Meanwhile, we are also conducting experiments on more complex scenarios (like learning facial expressions), hoping to deliver more insights in the future.

---

> ### Comment · Reviewer_BaC3 · 2024-11-25
>
> I have reviewed the authors response, their updates to the manuscript and comments of other reviewers.
>
> The authors make clearly stated assumptions as the basis for the model, which are validated on the datasets chosen (which now include a more "natural" dataset). While the assumption may well not hold for more complex datasets (as noted by several reviewers), I see this work as providing a useful, well-conducted basis for then considering more complex scenarios.
>
> I believe this work is of interest to the community and of publishable quality, so **I maintain my score**.

---

### Official Review · Reviewer_ipXQ · 2024-11-02

**Soundness:** 3
**Presentation:** 3
**Contribution:** 2
**Rating:** 6
**Confidence:** 4

**Summary:**

This paper proposes an unsupervised disentanglement algorithm between styles and content, based on the assumption that style information remains consistent within samples, but varies across samples; whereas content information varies within samples, but the aggregated content is stable across samples due to the fixed vocabulary.

**Strengths:**

1. This paper proposes an innovative disentanglement algorithm that does not rely on external labels of the style or content, making a novel contribution over the existing work, most of which requires external labels.

2. There is a clear rationale behind the proposed algorithm, and the rationale is interesting

3. The presentation in the paper is very clear.

**Weaknesses:**

1. Although the proposed method does not rely on any external labels, it seems to be built upon rather strong assumptions. In particular, the method assumes that the aggregated content of a sample should stay constant across different samples, at least more stable than the style. This assumption is very strong, and there are many scenarios where this may not hold. First, if the content information in each example is too small to exhaust all the possible combinations, then the content would still vary greatly across samples. For example, in the written digit case, if each example only contains one digit. Then the model may no longer be very successful. This limitation may also hold for many other scenarios, such as Chinese calligraphy, which typically only contains very few characters, paintings containing only one object, etc. Second, if the content information exhibits too much variability, then the content across different statements may still have more variability than style. For example, consider short passages of different languages, where the content is the meaning of the passage and the style is language. The content in this case still varies greatly across different passages. Meanwhile, the authors also acknowledge that the method cannot deal with large content vocabularies. This limitation, together with the limitation discussed above, significantly limits the applicability of the proposed method.

2. Related to the first limitation, the experiment section seems to focus on very simple, carefully crafted toy scenarios that happen to satisfy the assumptions of the proposed method. There is no experiment on real-world scenarios, such as disentangling real human speech, photos, paintings, music, passages, etc. The lack of real-world experiments adds to the concern that the assumptions made by the proposed method may deviate too much from real-world cases to render the method useful.

3. (Minor) On page 5, the authors state that 'Theoretically, we aim to measure the consistency of codebook usage distribution along the sample axis, which is not differentiable'. This is not necessarily true. We can derive a soft, differentiable distribution of codebook usage via a softmax codebook selection. That is, for each codebook selection, rather than have a one-hot vector indicating which code is selected, we can derive a softmax selection, which is very common in deep-learning-based vector quantization methods, such as VQ-VAE. As a result, I believe that using the mean content vector to represent the distribution information may be improved.

4. (Minor) In line 242, the authors state that 'Content should be more variable within samples than across samples'. This is an overly simple statement. A more precise version should be 'Content should be more variable within samples than the sample-wise aggregated content across samples'.

**Questions:**

Can the author elaborate a bit more on the experiment setting of Section 4.2?
'we evaluate the models’ content-style disentanglement ability by conducting a retrieval experiment to examine the nearest neighbors of every input zc and zs using ground truth content and style labels, evaluated by the area under the precision-recall curve (PR-AUC) and the best F1 score'

---

> ### Author Response · Authors · 2024-11-25
>
> We thank the reviewer for the detailed feedback and your interest in our work. Our responses are as follows:
>
> > W1: Although the proposed method does not rely on any external labels, it seems to be built upon rather strong assumptions. In particular, the method assumes that the aggregated content of a sample should stay constant across different samples, at least more stable than the style. This assumption is very strong.
>
> Allow me to re-explain why this assumption is domain-general and not strong at all -- there seems be a misunderstanding between "samples" and "distributions" in your argument. Your example of "each sample only containing one digit" is a possible outlier sample in the distribution, but it does not represent the majority and common cases. In fact, there could exist a lot of individual samples that yield high V3 losses, but it doesn't prevent V3 from learning content and style successfully, as there are almost no such skewed distributions in reality. If this is the case for a whole distribution, say, a human baby *always* sees one digit repeatedly written on paper at a time, he/she would probably think of digits as what we think of patterns on a plaid shirt -- content can now degrade to style.
>
> It is important to note that such "outlier" samples are indeed already present commonly in our training data (especially in SVHN where we can see numbers like "66" and "88"), but they bring no challenge for V3 during either training or inference. In fact, from the beginning of this project, we have been discussing possible domains/distributions that could fundamentally violate the general inductive bias of V3, but the only cases people could possibly imagine are collage-like artificial settings, e.g. "changing the instrument on every note", "changing the color on every letter". However, such setups deviates from the purpose of discovering the first-principle inductive bias of content-style disentanglement that aligns with genuine human perception developed in early ages.
>
> > W1: If the content information exhibits too much variability, then the content across different statements may still have more variability than style. For example, consider short passages of different languages, where the content is the meaning of the passage and the style is language.
>
> Again, there seems to be a misunderstanding between the "contents conveyed in individual samples", and the "content vocabulary shared by all samples".
> You are right that every passage talks about different things and the content can vary even more than language itself, but passages written in every language talk about same topics and similar subjects. Meanwhile, our method doesn't focus on learning the actual meaning of passages, but learning the underlying words themselves. We will clarify this more in the paper.
>
> > W2: The experiment section seems to focus on very simple, carefully crafted toy scenarios that happen to satisfy the assumptions of the proposed method.
>
> Thank you for pointing that out. We kindly refer you to the revised parts marked in blue in our latest revision, where we have tested V3 on Librispeech, which is a much more complex scenario given a larger number of content symbols and more segmentation noise. Despite the complexity of the new problem, V3 still shows better disentanglement ability and good content codebook interpretability compared to baselines.
>
> > W3: We can derive a soft, differentiable distribution of codebook usage via a softmax codebook selection.
>
> Thank you for your interest in the implementation details. V3 as a general inductive bias can have different instantiations. We know this is a possible implementation but didn't adopt it considering that it has a different dimension of variability, which mismatches with other terms in the V3 loss. Also, in our current experiments, we find that the current implementation has already shown good performance. Still, we would be happy to explore this implementation in future experiment.
>
> > W4: The authors state that 'Content should be more variable within samples than across samples'. This is an overly simple statement.
>
> Thank you for pointing this out. We have clarified this in the latest revision to prevent misunderstanding.
>
> > Q1: Can the author elaborate a bit more on the experiment setting of Section 4.2?
>
> Consider the content/style representation of a specific fragment as the anchor, we look for the nearest neighbors of it in all such representations in the test set. If a neighbor indeed belongs to the same content/style class, it's a true positive, otherwise a false positive. Same-classed fragments outside the neighbors are false negatives. We measure such precision and recall for every anchor, and then gradually increase the radius to plot the precision-recall curve.
>
> We hope that these can address your concerns, and would be thankful if you could reconsider the overall evaluation.

---

> > ### Comment · Reviewer_ipXQ · 2024-11-26
> > **Further questions on method's applicability**
> >
> > I would like to thank the authors for the detailed response.
> >
> > Regarding W1, perhaps the examples that I used were not adequate. I would like to ask a clarifying question: In your experiment of the written phone dataset, for example, what if each image only contains one digit? Could the method still work?

---

> ### Author Response · Authors · 2024-11-26
>
> Thank you for the follow-up question. No, V3 would not work and does not apply to the "one digit per image" scenario.
>
> As stated in the abstract, the goal is to learn content and style from a *sequence* of observations by leveraging their variabilities. "One digit per image" contradicts the assumption of "a sequence", making it impossible to observe these variabilities. (After all, we don't see all single-digit phone numbers in real life, so such cases are not considered.) That said, V3 works well when images with two digits are in the dataset (i.e., as long as there is an opportunity for the algorithm to observe variabilities), which indicates the robustness of V3. In the SVHN dataset, for example, where we see many images with one or two digits, V3 demonstrated pretty good performance.

---

> > ### Comment · Reviewer_ipXQ · 2024-11-27
> > **Following the discussion**
> >
> > Given this, can the authors further comment on the applicability of the proposed approach in the following cases (which are more realistic cases)?
> > 1.1. A set of calligraphy artworks where each sample only consists of one character;
> > 1.2. A set of art paintings where the content is a single animal (e.g. horse vs sheep vs cow), and the style is the color of the animal (e.g. white vs black vs yellow);
> > 2.1 Utterances of conversational speech where each small segment may contain multiple persons talking;
> > 2.2 Pieces of Concertos where each piece may contain alternating solos by different instruments.

---

> ### Author Response · Authors · 2024-11-27
>
> Thank you for providing these detailed examples. V3 is not applicable to cases 1.1 and 1.2. Images of single characters or single animals do not form sequences of observations. In fact, without prior knowledge of characters or animals, humans also struggle to distinguish between content and style. For example, we might perceive a black goat as a completely different kind of mammal rather than a variation of a goat. Similarly, we may classify animals of the same color as belonging to the same species, interpreting differences in shape merely as variations in looks. The same challenge arises with characters: without prior knowledge of scripts, it would be difficult to differentiate characters. In such cases, there are two varying factors, but it becomes unclear which represents content and which represents style unless prior domain knowledge is available. However, V3 is applicable if the dataset contains some images of more than one characters or some images of a group of animals, which is closer to what humans see in early ages.
>
> Both case 2.1 and 2.2 involve sequences of observations. For case 2.1, V3 is applicable if the speakers are alternating. The frequency of speaker alternation is always lower than the frequency of phoneme alternation, which aligns with the inductive bias of V3. V3 is applicable to case 2.2 for the same reason. However, V3 is not applicable to case 2.1 if the speakers are **always** speaking simultaneously, as this results in 'overlapping content,' which our current method cannot handle, as noted in the Limitations section. A potential solution would be to cascade V3 with a purely self-supervised source separation model. This model could first isolate independent sequences of observations before learning content and style. Addressing this challenge represents an interesting direction for future work.

---

> > ### Comment · Reviewer_ipXQ · 2024-11-27
> > **Score adjusted and further suggestions**
> >
> > The authors' response has largely resolved my concerns. I have adjusted the score accordingly.
> >
> > Since V3 is built on non-trivial assumptions, this paper would further benefit from a more systematic discussion of the assumptions or limitations. Either, it would be helpful to add the single content limitation (cases 1.1 and 1.2) to the limitation section, since the other limitations are already discussed there; or, even better, the authors could discuss the assumptions upfront before introducing the V3 constraints (Section 3.3). It would be a more natural flow to start from a set of assumptions (including multiple content sequences, pre-defined segments, non-overlapping content, etc.) and then discuss the constraints derived from the assumptions.

---

> ### Author Response · Authors · 2024-11-27
>
> Thank you for this valuable discussion and we are grateful for your reevaluation. Yes, we will take your advice when optimizing the writing flow! If you don't mind, we have some additional comments regarding the examples you mentioned above, as we find the comparison worthwhile.
>
> For case 1.1 and 1.2: There seems to be a gap between such cases and the natural human-learning scenarios. In a realistic scenario, calligraphy artworks almost always contain multiple characters (usually a short paragraph), and it is the variabilities manifested in the paragraphs (which V3 characterizes) that help humans to learn the concept of characters (content) and scripts (style). Similarly, we didn't learn animals and their colors from observing a single animal at one time when we were young -- we probably saw many animals in the wild and many animals in the zoo. By comparing the variabilities in different environments (similar to what V3 does), we may learn the animal species (content) and whether they are energetic or lazy (style). More precise learning would involve external supervision like labels or domain knowledge (System 2 thinking), which is beyond of scope of this study.
>
> For case 2.2: In practice, during training, we would randomly sample segments of contiguous fragments from the training audio. As a result, most sampled fragments would belong to the same instrument, which would bring little difference to the performance. It's like when we listen to the music, even if there are alternating instruments, our "attention" still parses a lot of their individual parts.
>
> It is worth noting that the task is more like unsupervised concept discovery from raw observations, rather than pattern recognition given an already well-defined content or style, so we cannot pre-define the styles to be the script/color/timbre. What we perceive as content and style are somewhat fluid, depending on how the data is formed and presented. V3 is trying to mirror this profound capability of human perception, where concepts like content and style *emerge* from natural observations.

---

### Official Review · Reviewer_oi3G · 2024-11-03

**Soundness:** 3
**Presentation:** 3
**Contribution:** 2
**Rating:** 6
**Confidence:** 2

**Summary:**

The paper proposes an unsupervised method that learns from raw observation and disentangles its latent space into content and style representations based on the meta-level insight of domain-general statistical differences between content and style.

**Strengths:**

1. The proposed method does not rely on external supervision such as labels.
2. The proposed method can generalize to unseen data.
3. The disentangled content representations align with human perceptions in some cases.

**Weaknesses:**

1. The paper does not explicitly discuss how the codebook size and codebook dimension are determined. This could limit the application of the proposed method to different datasets.
2. The experiments are all conducted on toy datasets. This weakens the method's practical use in real-world scenarios. For example, for audio disentanglement, the musical note disentanglement is too simple. It is recommended to perform real-world disentanglement experiments such as voice conversion using VCTK.

**Questions:**

See above.

---

> ### Author Response · Authors · 2024-11-25
>
> Thank you for your interest and for acknowledging our insight in this paper. Our responses are as follows:
>
> > W1: The paper does not explicitly discuss how the codebook size and codebook dimension are determined.
>
> Thank you for the suggestion, and we could certainly include some discussions on the choice of codebook size and dimension in the paper. Briefly, the codebook dimension is a hyperparameter much like the latent dimension of a Transformer -- it does not affect the performance too much as long as it is large enough to capture the needed information. It is always more secure to use a large dimension to avoid information loss, and in our experiments, we use 512 or 768, depending on the dimension of the input data itself. As for the codebook size, its value is set based on the kind of content vocabulary we want to learn -- with or without overspecification. In Appendix C.2, we show how different codebook sizes affect the interpretability of codebook entries -- if there is codebook redundancy, multiple entries will refer to one content -- but as long as the codebook size is larger than the number of content symbols, the performance on both interpretability and disentanglement remain good. Besides, we observe that if the codebook size is smaller than the number of content symbols (aka underspecified), the codebook collapses to poor interpretability.
>
> > W2: It is recommended to perform real-world disentanglement experiments.
>
> Thank you for the recommendation! Yes, we have conducted a new experiment and we kindly refer you to the added parts marked in blue in our latest revision. We have tested V3 on Librispeech, which is a much more complex scenario given the number of content symbols and the segmentation noise. Despite the complexity of the new problem, V3 still shows better disentanglement ability and good content codebook interpretability compared to baselines.

---

### Official Review · Reviewer_D5wf · 2024-11-04

**Soundness:** 2
**Presentation:** 3
**Contribution:** 2
**Rating:** 6
**Confidence:** 4

**Summary:**

This paper introduces an unsupervised method named V3 (variance-versus-invariance) for disentangling content and style representations from a sequence of observations. The method is based on the assumption that content varies significantly within a sample but maintains a consistent vocabulary across samples, while style remains relatively invariant within a sample but varies significantly across samples. V3 integrates this inductive bias into an encoder-decoder architecture and demonstrates its effectiveness across multiple domains and modalities, including music audio, handwritten digit images, and simple animations. The paper reports that V3 outperforms existing unsupervised methods and even surpasses supervised methods in out-of-distribution generalization and few-shot learning. Additionally, V3 exhibits symbolic-level interpretability, aligning machine representations closely with human knowledge.

**Strengths:**

1. The V3 method offers a novel unsupervised disentanglement approach that leverages the universal statistical differences between content and style. This is a creative application of variance-invariance constraints within an encoder-decoder framework, distinct from traditional methods that rely on supervision or paired data.
2. The loss calculation method proposed by V3, based on content and style, is concise and clear, aligning with human cognitive characteristics. Humans also tend to keep the style of the same sample similar while focusing on the content differences across different samples.
3. The model proposed in the paper is applicable to the learning of multiple modalities of style and content, including images, music, and animations, demonstrating strong generalizability. Moreover, the loss calculation pattern presented is not limited to the paper model but can also be applied to other related models for style and content learning.
4. The paper is well-structured, with a clear problem statement, methodology, and experimental evaluation. The use of datasets from multiple domains enhances the robustness of the research findings. The logic is strong, and the hierarchy is clear. The visualization of learned representations also helps readers better understand the results.

**Weaknesses:**

1. The paper's approach to learning sample content and style is reliant on datasets with a specific structure, which may limit its effectiveness when applied to images with varied or complex styles. For example, the Written Phone Numbers dataset contains only one digit in a specific position on each image, but the style of the digit can vary; in SPRITES, the character's position is fixed, and the form is also relatively uniform. If an image contains more content or more complex content, the model may have difficulties learning content and style.
2. The Ablation study reveals that certain components of the loss function may adversely affect performance metrics, suggesting a need for a deeper investigation into the roles and interactions of these components within V3.
3. It appears that V3 may have difficulty distinguishing between some similar contents during recognition. Similarly, although the paper may not mention it, V3 may also have similar problems with distinguishing styles, thus limiting the style discrimination between different samples.
4.Some important related works[a,b] are missing, which are also rely one some assumption to disentangle content and style.   The detailed comparison should be included, especially on the basic assumptions and the scope of suitable cases.
a. Retriever: Learning Content-Style Representation as a Token-Level Bipartite Graph
b. Rethinking Content and Style: Exploring Bias for Unsupervised Disentanglement.

**Questions:**

1. It is better to evaluate V3 on  more complex or challenging datasets, such as face datasets FFHQ? And It is better to add discussion on
 potential limitations of V3 when dealing with more complex data containing multiple contents or styles within a single sample.
2. Since V3 uses different losses to control content and style representations through self-supervised means, is there a risk that the representations intended to learn style might inadvertently capture content instead? How can it be ensured that there is not too much coupling between content and style representations? I suggest to conduct an analysis showing how much content information can be recovered from the style representations and vice versa.
3. Why is it necessary to apply VQ to content representations to form a codebook? What would be the impact on the learning results if VQ were not used? Seeing corresponding results in the Ablation study could be more convincing.
4. Are the numbers of content and style representations in V3 the same? Considering that the actual samples may have varying quantities suitable for representing content and style, would setting different numbers for content and style representations in the same training round make a significant difference in the results?

---

> ### Author Response · Authors · 2024-11-25
>
> Thank you for your interest in our work and the insightful comments. Our responses are as follows.
>
> > W1: If an image contains more content or more complex content, the model may have difficulties learning content and style.
>
> We consider segmentation as the prerequisite of learning content symbols, and this also holds in the learning process of humans. For example, it's because we can visually separate symbols written on paper unsupervisedly that we can learn systems of digits/letters/characters. Even if the segmentation is not perfect, which makes content more complex, V3 can still learn reasonable content and style information. (In the SVHN dataset, the bounding boxes of digits can be very noisy with tilts or parts of adjacent digits!) *Moreover, we have recently tested V3 on a large-scale speech dataset*, which is much more complex given the number of content symbols and the imperfect segmentation. We refer you to the added parts marked in blue in our latest revision. Despite the complexity of the new problem, V3 still shows better disentanglement ability and good content interpretability.
>
> > W2: The Ablation study reveals that certain components of the loss function may adversely affect performance metrics, suggesting a need for a deeper investigation.
>
> We thank the reviewer for this observation. The real variability scenario of content and style differs from dataset to dataset, and V3 is a domain-general approach that doesn't rely on any domain knowledge and still works well. On the one hand, the results also indicates that V3 is adaptable to different scenarios to achieve an even better performance compared to the universal setting. On the other hand, as it is difficult to know the best configuration for every dataset beforehand, the V3 constraints as a whole has already shown robust performance across domains.
>
> > W3: V3 may also have similar problems with distinguishing styles, thus limiting the style discrimination between different samples.
>
> Thank you for this insightful comment. You are totally right about the style ambiguity, but such ambiguity applies to human System 1 perception (which V3 tries to mirror) as well. For example, two composers or painters can have very similar styles in some specific periods of career, and it's hard to distinguish their works without deliberate study. What we often use to label "style", like the "name" of the artist or a "genre" of music, is usually an overly abstracted symbol compared to the rich and raw style we perceive. (But such lossy abstraction is unavoidable if we rely on human language, because the perceived style is too hard to describe.) As a System 1-learning model, V3 is designed to learn this raw form of content and style, and the ambiguities to humans before deliberate study are also ambiguous to V3.
>
> > W4: Related works
>
> Thank you for pointing out these interesting works! We have added them in the related work section in the latest revision. They both share a similar insight on content and style with V3, while focusing on different implementations and applications.
>
> > Q2: Is there a risk that the representations learn style might inadvertently capture content instead? How can it be ensured that there is not too much coupling?
>
> As the V3 loss is not symmetrical, there will be no risk of the content and style branches learning overturned information. Although the disentanglements are not always perfect, V3 still surpasses baselines as is shown in Table 1-6, as the content and style representations learned by V3 is good that their respective jobs but bad at the other's job.
>
> > Q3: Why is it necessary to apply VQ?
>
> Thank you for this interesting question. As stated in the introduction, we are interested in learning symbolized content. Intuitively, content is something that we could write down or record to convey to others so that they can understand and replay. If we think about the content as a language, there must be a tokenized vocabulary, which is what we are interested in learning. In other words, we use VQ not to improve the performance, but to characterize the nature of this vocabulary.
>
> > Q4: Are the numbers of content and style representations in V3 the same? Would setting different numbers make a significant difference?
>
> The numbers of content and style are not necessarily the same. None of our experiments have the same number of content and style representations. In fact, there is not a number of styles, as style is naturally continuous (hope the example of composers/composition styles above can help). We rely on the style classes only for evaluation purposes and connection to specific tasks, but do not rely on them in training. In SVHN where every sample has a completely different style (and even digits on the same image can have different styles), V3 still works well in learning the content and style information.
>
> We hope that these can address your concerns and would be grateful if you could reconsider the overall evaluation.

---

> > ### Comment · Reviewer_D5wf · 2024-11-28
> >
> > Thanks for the response. My main concern is still on the assumption and limitation of the methods, which has been roughly resolved, but I would suggest to provide more clear discussion and experiment on what kind of data that the proposed method can work or not. I have increased my score.

---

> ### Author Response · Authors · 2024-11-28
>
> Thank you for your thoughtful feedback and for increasing your score. We appreciate your suggestion to further clarify the types of data to which our method is applicable or not. Your input is invaluable in improving our work.

---

### Official Review · Reviewer_GuWP · 2024-11-04

**Soundness:** 3
**Presentation:** 3
**Contribution:** 3
**Rating:** 6
**Confidence:** 2

**Summary:**

They propose an unsupervised method to disentangle content and style representations.  They use vector-quantized autoencoder architecture and incorporate variance-versus-invariance constraints. Their method benefits from meta-level inductive bias, and shows out-of-distribution generalization. They demonstrate their method’s effectiveness across diverse applications such audio, images and video clips. Moreover, the learned codebook enables interpretable features.

**Strengths:**

++ Unsupervised approach for separating style and content, which is a plus for domains where labeled data is unavailable or scarce

++ Generalizes across audio, image and video data.

++ Content representations show interpretable properties.

++ Performs well for out of distribution styles

**Weaknesses:**

-- They assume content elements are distinct, which can limit its application in domains with overlapping content.

-- The method is optimized for data with predefined fragments, which may not generalize well to continuous or unsegmented data types.

-- The experiments are done on relatively small datasets.

**Questions:**

- How does the choice of codebook size (K) affect interpretability and disentanglement performance?

- The method uses specific values for the relativity parameter r and the V3 loss weight beta  depending on the dataset. Could you clarify the intuition behind selecting these values? How sensitive the performance is to different values of these hyperparameters?

- The paper mainly utilizes small datasets with specific content-style boundaries. How does it scale with larger datasets?

Minor issues:
Line 463: The term "contennt" should be corrected to "content."

---

> ### Author Response · Authors · 2024-11-25
>
> Thank you for the detailed feedback, and we are grateful for your interest in the deeper analysis of our work. Our responses are as follows.
>
> > Weaknesses in terms of dataset size, pre-defined fragments, and distinct (non-overlapping) contents.
>
> We have conducted a new experiment, testing V3 on a *large-scale* speech dataset, covering one more domain as well as a much more complex scenario. Please see the added parts in our latest revision (marked in blue). The new experiment shows that V3 consistently achieves better disentanglement and content codebook interpretability compared to the baselines, despite the complexity of the new problem.
>
> Also, thank you for pointing out the limitations of distinct content and predefined fragments. As stated in our Limitation section, these are indeed cases our current method cannot handle, and we regard them as future works -- e.g., how to learn the concept of pitch and timbre from arbitrary polyphonic music audio where the notes are overlapped. Nevertheless, we see the current setting of V3 as fundamental enough, as it mirrors how concepts of content and style emerge in human system-1 perception during early ages. In contrast, learning overlapping contents can be seen as a case of System-2 learning, which involves knowledge-based reasoning  -- e.g., a child first learns to perceive individual note pitches via System-1 and later on learns to derive the concept of "chord" (a combination of pitches) with System-2. As for the segmentation problem, other self-supervised learning modules (e.g. self-supervised segmentation like DINO) can be combined with V3 and trained end-to-end.
>
> > Q1: How does the choice of codebook size (K) affect interpretability and disentanglement performance?
>
> Table 1 and 2 compares the disentanglement abilities when using different K values, and Table 6 shows the interpretability difference. In our experiments, the K value doesn't affect the disentanglement ability too much, while changing how the codebook is interpreted. Choosing a K value with no codebook redundancy (e.g. K=10 in learning digits) will result in a one-to-one alignment from codebook entries to digits, which is a precise and concise vocabulary. If the K value is larger (e.g. K=40 in learning digits), more than one codebook entries will refer to one content, just like in natural language, multiple words can simultaneously mean one thing/object/item/entity. Appendix C.2 would be an intuitive demonstration.
>
> > Q2: Could you clarify the intuition behind selecting these values? How sensitive the performance is to different values of these hyperparameters?
>
> Tuning these parameters is as simple as tuning the learning rate. In our experiments the r value ranges from 5 to 15. The more complex the dataset, the smaller r should be. It's recommended to start from a smaller value and gradually increase it if a small r is too easy for the model, but we do not need to find a maximum r to get good performance. We recommend setting beta so that the reweighted reconstruction loss and V3 loss are in the same order of magnitude after the first few steps. This enables the whole training process to have three phases: 1) learning to roughly reconstruct; 2) learning content and style; 3) learning to refine reconstruction.
>
> > Q3: The paper mainly utilizes small datasets with specific content-style boundaries. How does it scale with larger datasets?
>
> Would you mind clarifying what you mean by "specific content-style boundaries"?
>
> If you mean the "boundary" between content and style in human perception, we argue that in most domains, if not all, human perceive certain information conveyed as the content, and the leftover as style. For example, the symbolic language is the content in human speech, and the leftover style is intonation, prosody, timbre and etc. (Please refer to our introduction for more examples.) The boundary between content and style is an innate nature of human perception, which does not depend on the size of the dataset.
>
> If you mean the "boundaries of fragments", self-supervised segmentation models can be easily cascaded with our approach, which we leave for future work as segmentation itself is not the focus of this work. And even if the segmentations are not perfect (e.g. in SVHN, the bounding boxes of digits can be very noisy -- digits are not always centered, and the box can even contain parts of adjacent digits!), V3 can still learn reasonable content and style information.
>
> > The term "contennt" should be corrected to "content."
>
> Thank you for the sharp eyes! It has been fixed in the latest version.
>
> We hope these responses are helpful and would be happy to provide further clarifications if needed.

---

> > ### Comment · Reviewer_GuWP · 2024-12-01
> >
> > Thanks to the reviewers for addressing my concerns, I believe this paper can be a good contribution to the community, therefore I maintain my original score of 6.

---

### Meta-Review · Area_Chair_FKJU · 2024-12-20

**Metareview:**

This paper introduces V3 (variance-versus-invariance), an unsupervised method designed to effectively disentangle content and style representations from sequences of observations without relying on domain-specific labels or knowledge.

The reviewers acknowledged the novelty and contribution of this work for the V3 method's ability to separate style and content without relying on external labels or paired data, making it highly advantageous for domains with scarce or unavailable labeled data (Reviewer GuWP and ipXQ). In addition, reviewers highlighted V3's robust performance across diverse data types, including audio, images, and video (Reviewer GuWP and D5wf).

Some reviewers also raised concerns that V3 assumes content elements are distinct and consistent across samples (Reviewer GuWP and ipXQ). Moreover, it was noted that V3 is optimized for data with predefined fragments, potentially limiting its performance on continuous or unsegmented data types (Reviewer GuWP and D5wf). Last but not least, almost all the reviewers mentioned the reliance on relatively small or toy datasets, such as handwritten digits and simple animations, which raises concerns about the method's practical applicability in real-world scenarios.

During the rebuttal, the authors made great efforts to address the reviewers’ concerns. The main concern about the assumption and lacking experiments on a larger dataset, had been well addressed by the additional clarifications provided by the authors. Some other minor concerns are acknowledged by the authors and claimed to be regarded in future work. Considering all factors, the AC therefore recommends acceptance of this paper.

**Additional Comments On Reviewer Discussion:**

Reviewers GuWP and ipXQ highlighted concerns that V3 assumes content elements are distinct and maintain stability across samples, and most reviewers mentioned the reliance on relatively small or toy datasets. In the rebuttal period, the authors added a comprehensive clarification to answer the concerns and questions the reviewers proposed. Reviewers GuWP and D5wf noted that V3 is optimized for data with predefined fragments. The authors acknowledged these limitations and clarified some of these to be regarded for future work. Some other minor concerns the reviewers mentioned are well addressed by the authors. Taking all points into account, the Area Chair recommends acceptance of this paper.

---

### Decision · Program_Chairs · 2025-01-22

Accept (Poster)